# TimeAutoDiff: Generation of Heterogeneous Time Series data via Latent Diffusion Model

## Abstract

In this paper, we leverage the power of latent diffusion models to generate synthetic time series tabular data. Along with the temporal and feature correlations, the heterogeneous nature of the feature in the table has been one of the main obstacles in time series tabular data modeling. We tackle this problem by combining the ideas of the variational auto-encoder (VAE) and the denoising diffusion probabilistic model (DDPM). Our model named as `TimeAutoDiff` has several key advantages including (1) *Generality*: the ability to handle the broad spectrum of time series tabular data with heterogeneous, continuous only, or categorical only features; (2) *Fast sampling speed*: entire time series data generation as opposed to the sequential data sampling schemes implemented in the existing diffusion-based models, eventually leading to significant improvements in sampling speed, (3) *Time varying metadata conditional generation*: the implementation of time series tabular data generation of heterogeneous outputs conditioned on heterogenous, time varying features, enabling scenario exploration across multiple scientific and engineering domains. (4) *Good fidelity and utility guarantees*: numerical experiments on eight publicly available datasets demonstrating significant improvements over state-of-the-art models in generating time series tabular data, across four metrics measuring fidelity and utility; Codes for model implementations are available at the supplementary materials.

## 1 Introduction

Synthesizing tabular data is crucial for data sharing and model training. In the healthcare domain, synthetic data enables the safe sharing of realistic but non-sensitive datasets, preserving patient confidentiality while supporting research and software testing (Yoon et al., 2023). In fields like fraud detection (Padhi et al., 2021b; Hsieh et al., 2024; Cheng et al., 2024), where anomalous events are rare, synthetic data can provide additional examples to train more effective detection models. Synthetic datasets are also vital for scenario exploration, missing data imputation (Tashiro et al., 2021; Ouyang et al., 2023), and practical data analysis experiences across various domains.

Given the importance of synthesizing tabular data, many researchers have put enormous efforts into building tabular synthesizers with high fidelity and utility guarantees. For example, CTGAN (Xu et al., 2019) and its variants (Zhao et al., 2021; 2022) (e.g., CTABGAN, CTABGAN+) have gained popularity for generating tabular data using Generative Adversarial Networks (Goodfellow et al., 2020) (GANs). Recently, diffusion-based tabular synthesizers, like Stasy (Kim et al., 2022), have shown promise, outperforming GAN-based methods in various tasks. Yet, diffusion models (Ho et al., 2020; Song et al., 2020b) were not initially designed for heterogeneous features. New approaches, such as those using Doob's h-transform (Liu et al., 2022), TabDDPM (Kotelnikov et al., 2022), and CoDi (Lee et al., 2023), aim to address this challenge by combining different diffusion models (Song et al., 2020b; Hoogeboom et al., 2022) or leveraging contrastive learning (Schroff et al., 2015) to co-evolve models for improved performance on heterogeneous data. Most recently, researchers have used the idea of a latent diffusion model, i.e., AutoDiff (Suh et al., 2023) and TabSyn (Zhang et al., 2023a), to model the heterogeneous features in tables and prove its empirical effectiveness in various tabular generation tasks.

However, the tabular synthesizers mentioned above focus solely on generating tables with independent and identically distributed (*i.i.d.*) rows. They face difficulties in simulating time series tabular data

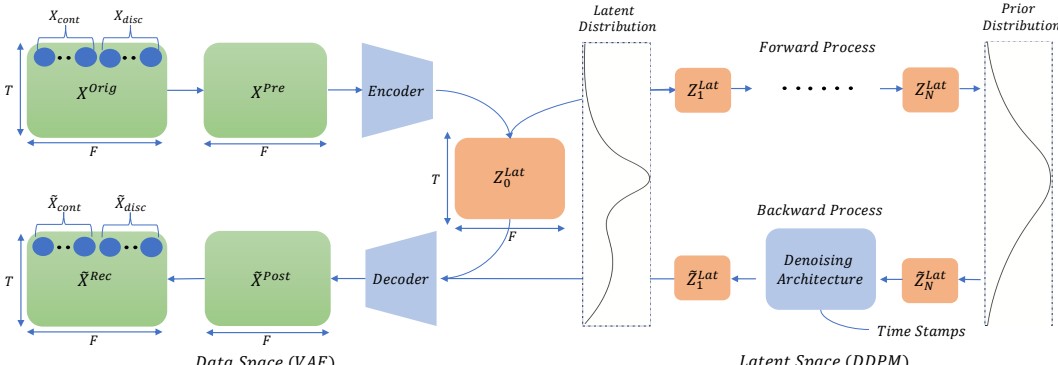

Figure 1: The overview of `TimeAutoDiff`: the model has three components: (1) pre- and post-processing steps for the original and synthesized data; (2) VAE for training encoder and decoder, and for projecting the pre-processed data to the latent space; (3) Diffusion model for learning the distribution of projected data in latent space and generating new latent data. Note that the dimension of the latent matrix $Z_0^{\text{Lat}} \in \mathbb{R}^{T \times F}$ is set to be the same as that of the original data.

due to the significant inter-dependences among features and the intricate temporal dependencies that unfold over time. In this paper, motivated from (Suh et al., 2023; Zhang et al., 2023a), we propose a new model named `TimeAutoDiff`, which combines Variational Auto-encoder (VAE) (Kingma & Welling, 2013) and Denoising Diffusion Probabilistic Model (DDPM) (Ho et al., 2020) to tackle the above challenges in time series tabular modeling. In the remainder of this section, we define our problem formulation, introduce motivations of TimeAutoDiff, outline the contributions of our work, and review relevant literature to establish the context of our paper.

### 1.1 PROBLEM FORMULATION, MOTIVATION, AND CONTRIBUTIONS

**Problem Formulation.** Our goal is to learn the joint distribution of time series tabular data of a $T$-sequence $(\mathbf{x}_1, \dots, \mathbf{x}_T)$. Each observation $\mathbf{x}_j$, where $\mathbf{x}_j := [\mathbf{x}_{\text{Cont},j}, \mathbf{x}_{\text{Disc},j}]$ is an $F$-dimensional feature vector that includes both continuous ($\mathbf{x}_{\text{Cont},j}$) and discrete variables ($\mathbf{x}_{\text{Disc},j}$), reflecting the heterogeneous nature of the dataset. Throughout this paper, we assume there are $B$ i.i.d. observed sequences sampled from $\mathbb{P}(\mathbf{x}_1, \dots, \mathbf{x}_T)$. We additionally assume that each record in the time-series tabular data includes a timestamp, formatted as 'YEAR-MONTH-DATE-HOURS'. This timestamp serves as an auxiliary variable to aid an training / inference step in `TimeAutoDiff`, which will be detailed shortly. The overview of `TimeAutoDiff` is provided in Figure 1.

**Motivation of TimeAutoDiff.** The main motivation for combining the two models, VAE and DDPM, is to accurately capture the distribution of heterogeneous features in the data. Diffusion model has recently gained a lot of attentions in a time series community, because of its ability generating complex and high quality sequences. (See Lin et al. (2024); Yang et al. (2024b) and references therein for more detailed reasons.) Nonetheless, current literatures only focus on modeling continuous time series data. This is mainly attributed to the fact that the diffusion model is originally designed for capturing distributions on continuous space. In our work, to deal with heterogeneous features, the $\beta$-VAE (Higgins et al., 2017) is employed for projecting the time series data to continuous latent space. The autoencoder framework has been widely employed in tabular data modeling to address heterogeneity, leveraging the reconstruction error in its objective function (Desai et al., 2021; Suh et al., 2023; Zhang et al., 2023a). Inspired from this observation, we combine these two models for modeling a time-series data with heterogeneous features. Furthermore, dependencies along temporal and feature dimensions can be captured through the sophisticated architectural designs of VAE and DDPM denoiser. Specifically, in both models, we use the inductive bias of Recurrent Neural Network (RNN) (Hochreiter & Schmidhuber, 1997) and Bi-directional RNN (Bi-RNN) (Schuster & Paliwal, 1997) to capture the temporal dependences of sequences. Our unique design of DDPM denoiser captures the feature dependences. More details are provided in Section 2.

***Contribution 1.*** *Sampling time* for new data sequence generation is significantly reduced compared to other SOTA diffusion-based time series models like TSGM (Lim et al., 2023) and diffusion-ts (Yuan & Qiao, 2023), which rely on sequential sampling. Existing synthesizers typically model the conditional distribution $\mathbb{P}(\mathbf{x}_t | \mathbf{x}_{t-1}, \dots, \mathbf{x}_1)$ and generate $\mathbf{x}_t$ sequentially for $t \in \{2, \dots, T\}$.

In contrast, our model learns the entire distribution $\mathbb{P}(\mathbf{x}_T, \mathbf{x}_{T-1}, \ldots, \mathbf{x}_1)$ and generates the whole sequence at once. The difference in sampling times becomes more pronounced as $T$ increases since diffusion models require multiple denoising steps for each sample. For verifications, our model generates long sequential data (i.e., $T = 900$ in Appendix L), while other diffusion baseline methods suffer from generating much shorter sequential data (i.e., $T = 24$ in Table 1). Additionally, our approach avoids the accumulating errors commonly associated with sequential sampling.

**_Contribution 2._** `TimeAutoDiff` accommodates _conditional generation_. The model can be conditioned on heterogeneous sequential metadata. [1] Given $B$ i.i.d. pairs of multivariate time series $\mathbf{x}_i$ and time-varying metadata $\mathbf{c}_i$ (i.e., $\mathcal{D}_{x,c} = \{(\mathbf{x}_i, \mathbf{c}_i)\}_{i=1}^{B}$), our model learns the conditional distribution $p(\mathbf{x}|\mathbf{c})$. Notably, both $\mathbf{x}_i$ and $\mathbf{c}_i$ can represent _multivariate heterogeneous_ and _sequential_ data. Additionally, static variables (e.g., gender, ethnicity) can also be incorporated as conditions $\mathbf{c}$ in our model. This capability unlocks significant potential for the model to be employed in counterfactual scenario exploration across diverse scientific and engineering domains. We demonstrate this potential through two specific examples under synthetic and real-world (Traffic dataset) settings in Section 4.3.

**_Contribution 3._** _Numerical comparisons_ of `TimeAutoDiff` with other models (with publicly available codes), namely, TimeGAN (Yoon et al., 2019), Diffusion-ts (Yuan & Qiao, 2023), TSGM (Lim et al., 2023), CPAR (Zhang et al., 2022), and DoppelGANger (Lin et al., 2020) are conducted comprehensively across eight real-world datasets under various metrics. (See Appendix C for descriptions of the datasets.) Specifically, for measuring the fidelities of temporal correlations between synthetic and real heterogeneous timeseries tabular data, we develop a new metric, named **_Temporal Discriminative Score_**. Inspired from the paper (Yoon et al., 2019; Zhang et al., 2022), this metric computes discriminative scores (Yoon et al., 2019) of distributions of inter-row differences (Zhang et al., 2022) in generated and original sequential data.

## 1.2 RELEVANT LITERATURE

To our knowledge, not many models in literature can deal with time series tabular data with a heterogeneous nature. We categorize the incomplete list of existing models into three parts: (1) GAN-based models, (2) Diffusion-based models, and (3) GPT-based / Parametric models.

**GAN-based models.** TimeGAN (Yoon et al., 2019) is one of the most popular time series data synthesizers based on the GAN framework. Notably, they used the idea of latent GAN employing the auto-encoder for projecting the time series data to latent space and model the distribution of the data in latent space through the GAN framework. Recently proposed Electric Health Record (in short EHR)-Safe (Yoon et al., 2023) integrates a GAN with an encoder-decoder module to generate realistic time series and static variables in EHRs. EHR-M-GAN (Li et al., 2023) employs distinct encoders for each data type, enhancing the generation of mixed-type time series in EHRs. Despite these advancements, GAN-based methods still encounter challenges such as non-convergence, mode collapse, generator-discriminator imbalance, and sensitivity to hyperparameter selection, underscoring the need for ongoing refinement in time series data synthesis.

**Diffusion-based models.** Most recently, TimeDiff (Tian et al., 2023) adopts the idea from TabDDPM combining the multinomial and Gaussian diffusion models to generate a synthetic EHR time series tabular dataset. DPM-EHR (Kuo et al., 2023) suggested another diffusion-based mixed-typed EHR time series synthesizer, which mainly relies on Gaussian diffusion and U-net architecture. TSGM (Lim et al., 2023) used the idea of the latent conditional score-based diffusion model to generate continuous time series data. However, TSGM is highly overparameterized and its training, inference, and sampling steps are quite slow. Diffusion-TS (Yuan & Qiao, 2023) takes advantage of the latent diffusion model employing transformer-based auto-encoder to capture the temporal dynamics of complicated time series data. Specifically, they decompose the seasonal-trend components in time series data making the generated data highly interpretable. One important model in the literature, CSDI (Tashiro et al., 2021), uses a 2D-attention-based conditional diffusion model to impute the missing continuous time series data.

---

[1] During the preparation of this manuscript, `TimeWeaver` (Narasimhan et al., 2024) was introduced in the literature. While it is also designed for time-varying metadata conditional generation, it focuses solely on the conditional generation of continuous outputs, and its code is not publicly available yet.

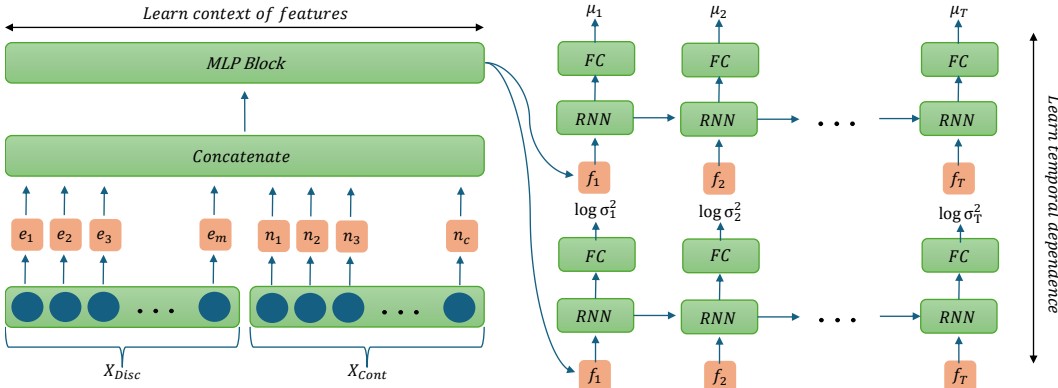

Figure 2: The schematic architecture of the encoder in VAE. The encoder has three main parts: (1) encoding of heterogeneous features having both discrete and continuous data; (2) learning the correlations of features through MLP block; (3) learning the temporal dependence through two RNNs.

**GPT-based / Parametric models.** TabGPT (Padhi et al., 2021b) is a GPT2-based tabular data synthesizer, which can deal with both single and multi-sequence mixed-type time series datasets. Data generation of TabGPT is performed by first inputting initial rows of data, then generating synthetic rows based on the context of previous rows. CPAR (Zhang et al., 2022) is an autoregressive model designed for synthesizing multi-sequence tabular data, i.e., sequences from multiple entities in one table. They use different parametric models (i.e., Gaussian, Negative Binomial) for modeling different datatypes (i.e., continuous, discrete). However, independent parametric design of each feature ignores the correlations among features.

## 2 PROPOSED MODEL: TIMEAUTODIFF

In this section, the constructions on variational auto-encoder (VAE) and diffusion models are provided. The pre- and post-processing steps of data are deferred in the Appendix D.

**Encoder in VAE:** The pre-processed input data $\mathbf{x}^{\text{Proc}} := [\mathbf{x}^{\text{Proc}}_{\text{Disc}}; \mathbf{x}^{\text{Proc}}_{\text{Cont}}] \in \mathbb{R}^{B \times T \times F}$ is fed into the VAE. The architecture of the encoder is illustrated in Figure 2. Motivated by TabTransformer (Huang et al., 2020), we encode the discrete feature $\mathbf{x}_j \in \mathbf{x}^{\text{Proc}}_{\text{Disc}}$ with $j \in \{1, \dots, m\}$ into a $d$-dimensional (where $d$ is consistently set at 128 in this paper) continuous representation. This is achieved using a lookup table $\mathbf{e}(\cdot) \in \mathbb{R}^d$ with $m$ representing the total number of discrete features. The goal of introducing embedding for the discrete variables is to allow the model to differentiate the classes in one column from those in the other columns. To embed the continuous features, we employ a frequency-based representation. Let $\nu$ be a scalar value of the $i$-th continuous feature in $\mathbf{x}^{\text{Proc}}_{\text{Cont}} \in \mathbb{R}^{B \times T \times c}$. Similar to (Luetto et al., 2023), $\nu$ is projected to the embedding spaces as follows:

$$n_i(\nu) := \text{Linear}\big(\text{SiLU}\big(\text{Linear}\big([\sin(2^0 \pi \nu), \cos(2^0 \pi \nu), \cdots, \sin(2^7 \pi \nu), \cos(2^7 \pi \nu)]\big)\big)\big) \in \mathbb{R}^d. \quad (1)$$

The embedding dimensions of discrete and continuous features are set to be the same as $d$ for simplicity. The sinusoidal embedding in equation 1 plays a crucial role in reconstructing the heterogeneous features. Our empirical observations indicate that omitting this embedding degrades the reconstruction fidelity of continuous features compared to their discrete counterparts, which will be verified in the ablation test in the following section. We conjecture this is attributed to the fact that deep networks are biased towards learning the low-frequency functions (i.e., spectral bias (Rahaman et al., 2019)), while the values in continuous time series features often have higher frequency variations.

The embedded vectors $e_1, \dots, e_m, n_1, \dots, n_c$ of each row in the input data are concatenated into a vector of dimension $(m + c)d$ and are inputted to the MLP block. The output tensor from the MLP block, denoted as $[f_1^{(i)}, f_2^{(i)}, \dots, f_T^{(i)}]_{i=1}^B \in \mathbb{R}^{B \times T \times F}$, is fed to two separate RNNs for modeling the mean and covariance of the latent distribution. Each RNN is unfolded over $T$ time horizons, and the vectors $\{f_j\}_{j=1}^T$ are fed to each network to capture the temporal dependencies of the input data. Henceforth, we omit the notation for the batch index when it is clear from context. The two RNNs' outputs are $\mu := [\mu_1, \mu_2, \dots, \mu_T]^T \in \mathbb{R}^{T \times F}$ and $\log \sigma^2 := [\log \sigma_1^2, \log \sigma_2^2, \dots, \log \sigma_T^2]^T \in \mathbb{R}^{T \times F}$,

Figure 3: The schematic architecture of the $\epsilon_\theta(\mathbf{Z}_n^{\text{Lat}}, n, \mathbf{t}, \mathbf{ts})$ in diffusion model. The inputs to the architecture $\epsilon_\theta$ include the noisy latent matrix $\mathbf{Z}_n^{\text{Lat}}$ at the $n$th diffusion step, the diffusion step $n$, the normalized time points $\mathbf{t}$, and the *time stamps* ($\mathbf{ts}$). The embedded inputs are processed through a Bi-directional RNN, which captures temporal dependencies in both forward and backward directions.

respectively. Then, we have a latent embedding tensor $\mathbf{Z}^{\text{Lat}} := \mu + \mathbf{E} \odot \sigma \in \mathbb{R}^{T \times F}$, where each entry in $\mathbf{E} \in \mathbb{R}^{T \times F}$ is from standard Gaussian distribution, and $\odot$ denotes an element-wise multiplication. The size of $\mathbf{Z}^{\text{Lat}}$ is set to be the same as that of the input tensor through fully-connected (FC) layers topped on the outputs of two RNNs.

**Dencoder in VAE:** We found a simple MLP block and linear layers work well as a decoder of VAE. First, we apply MLP to $\mathbf{Z}^{\text{Lat}}$ as in equation 2:

$$\text{MLPBlock}(\mathbf{x}) := \text{Linear}(\text{ReLU}(\text{Linear}(\mathbf{x}))), \quad \mathbf{x}^{\text{Pre-Out}} = \text{MLPBlock}(\mathbf{Z}^{\text{Lat}}). \tag{2}$$

Additional linear layers are applied to $\mathbf{x}^{\text{Pre-Out}}$ with separate layers, designed for each of the data types; that is, $\mathbf{x}_{\text{Cont}}^{\text{Out}} := \text{Sigmoid}(\text{Linear}(\mathbf{x}^{\text{Pre-Out}}))$, $\mathbf{x}_{\text{Bin}}^{\text{Out}} := \text{Linear}(\mathbf{x}^{\text{Pre-Out}})$, and $\mathbf{x}_{\text{Cate}}^{\text{Out}} := \text{Linear}(\mathbf{x}^{\text{Pre-Out}})$, denoting continuous, binary, and categorical outputs of the decoder. Here, we divide the discrete variables into two groups: $[\mathbf{x}_{\text{Bin}}, \mathbf{x}_{\text{Cate}}]$, where $\mathbf{x}_{\text{Bin}}$ represents binary variables, and $\mathbf{x}_{\text{Cate}}$ represents categorical variables with more than two labels. For numerical features, a sigmoid activation function scales the outputs to $[0, 1]$, matching the pre-processed input. The dimensions of $\mathbf{x}_{\text{Cont}}^{\text{Out}}$ and $\mathbf{x}_{\text{Bin}}^{\text{Out}}$ match their respective inputs, while $\mathbf{x}_{\text{Disc}}^{\text{Out}}$ has a dimension of $\sum_i K_i$, where $K_i$ is the number of categories in each categorical variable. Output dimensions are set to align with the requirements of MSE, BCE, and CE losses in PyTorch. The decoder structure is provided in Appendix L.

**Obj. function & Training of VAE:** The reconstruction error in the VAE is defined as the sum of mean-squared error (MSE), binary cross entropy (BCE), and cross-entropy (CE) between the input tuple $[\mathbf{x}_{\text{Bin}}^{\text{Proc}}, \mathbf{x}_{\text{Cate}}^{\text{Proc}}, \mathbf{x}_{\text{Cont}}^{\text{Proc}}]$ and the output tuple from decoder $[\mathbf{x}_{\text{Bin}}^{\text{Out}}, \mathbf{x}_{\text{Cate}}^{\text{Out}}, \mathbf{x}_{\text{Cont}}^{\text{Out}}]$:

$$\ell_{\text{recons}}(\mathbf{x}^{\text{Proc}}, \mathbf{x}^{\text{Out}}) = \text{BCE}(\mathbf{x}_{\text{Bin}}^{\text{Proc}}, \mathbf{x}_{\text{Bin}}^{\text{Out}}) + \text{CE}(\mathbf{x}_{\text{Disc}}^{\text{Proc}}, \mathbf{x}_{\text{Disc}}^{\text{Out}}) + \text{MSE}(\mathbf{x}_{\text{Cont}}^{\text{Proc}}, \mathbf{x}_{\text{Cont}}^{\text{Out}}). \tag{3}$$

Following (Zhang et al., 2023a), we use $\beta$-VAE (Higgins et al., 2017) instead of ELBO loss, where a coefficient $\beta (\geq 0)$ balances between the reconstruction error and KL-divergence of $\mathcal{N}(0, \mathcal{I}_{TF \times TF})$ ($\mathcal{I}_{TF \times TF}$ denotes an identity matrix of dimension $\mathbb{R}^{TF \times TF}$) and $\mathbf{Z}^{\text{Lat}} \sim \mathcal{N}(\text{vec}(\mu), \text{diag}(\text{vec}(\sigma^2))$. The notations $\text{vec}(\cdot)$ and $\text{diag}(\cdot)$ are vectorization of input matrix and diagonalization of input vector, respectively. Finally, we minimize the following objective function $\mathcal{L}_{\text{Auto}}$ for training VAE:

$$\mathcal{L}_{\text{Auto}} := \ell_{\text{recons}}(\mathbf{x}^{\text{Proc}}, \mathbf{x}^{\text{Out}}) + \beta \mathcal{D}_{\textbf{KL}}\big(\mathcal{N}(\text{vec}(\mu), \text{diag}(\text{vec}(\sigma^2))) \, || \, \mathcal{N}(0, \mathcal{I}_{TF \times TF})\big). \tag{4}$$

Similar to (Zhang et al., 2023a), our model does not require the distribution of embeddings $\mathbf{Z}^{\text{Lat}}$ to follow a standard normal distribution strictly, as the diffusion model additionally handles the distributional modeling in the latent space. Following Zhang et al. (2023a), we adopt the adaptive schedules of $\beta$ with its maximum value set as $0.1$ and minimum as $10^{-5}$, decreasing the $\beta$ by a factor of $0.7$ (i.e., $\beta^{\text{new}} = 0.7\beta^{\text{old}}$) from maximum to minimum whenever $\ell_{\text{recons}}$ fails to decrease for a predefined number of epochs. The effects of $\beta$-scheduling will be more detailed in Section 4.

**Diffusion for Time Series:** `TimeAutoDiff` is designed to generate the entire time series at once, taking the data of shape $T \times F$ as an input. This should be contrasted to generating rows in the table sequentially (i.e., for instance, Lim et al. (2023)). We extend the idea of DDPM to make it accommodate the time series data of shape $T \times F$ at one time. For readers' convenience, we provide the framework of DDPM in the Appendix F.

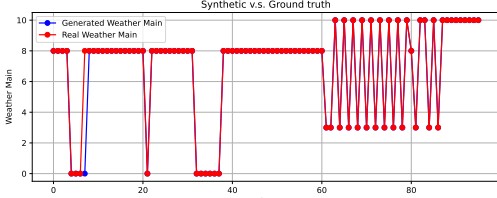 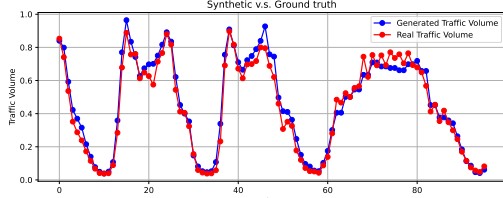

Figure 4: In Traffic dataset, 'Weather Main' (categorical, textual weather description) and 'Traffic Volume' (continuous, hourly traffic on westbound I-94, Minneapolis-St. Paul) are generated conditionally on remaining variables over 96 hourly timestamps (i.e., $T = 96$). The generated data shows great fidelity to the real data. (Both variables are pre-processed. See Appendix D for more details)

Let $\mathbf{Z}_0^{\text{Lat}} \in \mathbb{R}^{T \times F}$ denote the input latent matrix from VAE and let $\mathbf{Z}_n^{\text{Lat}} := [\mathbf{z}_{n,1}^{\text{Lat}}, \mathbf{z}_{n,2}^{\text{Lat}}, \ldots, \mathbf{z}_{n,F}^{\text{Lat}}]$ be the noisy matrix after $n \in \{1, 2, \ldots, N\}$ diffusion steps, where $\mathbf{z}_{n,j}^{\text{Lat}} \in \mathbb{R}^T$ is the $j$-th column of $\mathbf{Z}_n^{\text{Lat}}$. The perturbation kernel $q(\mathbf{z}_{n,j}^{\text{Lat}}|\mathbf{z}_{0,j}^{\text{Lat}}) = \mathcal{N}(\sqrt{\bar{\alpha}_n}\mathbf{z}_{0,j}^{\text{Lat}}, (1-\bar{\alpha}_n)\mathcal{I}_{T \times T})$ is applied independently to each column $\mathbf{z}_{0,j}^{\text{Lat}}$ $j \in [F]$, where $\bar{\alpha}_n := \Pi_{i=1}^n \alpha_i$ with $\{\alpha_i\}_{i=1}^n \in [0, 1]^n$ being a decreasing sequence over $i$. (We use the linear noise scheduling from DDPM. Refer to the Appendix K for details.) Here, we treat each column of $\mathbf{Z}_0^{\text{Lat}}$ as a discretized measurement of univariate time series function in the latent space, adding noises independently. But this does not mean we do not model the correlations along the feature dimension in $\mathbf{Z}_0^{\text{Lat}}$ (Biloš et al., 2023). The reverse process for sampling takes an entire latent matrix and captures these correlations. A similar idea has been used in TabDDPM (Kotelnikov et al., 2022) for modeling categorical variables. Under this setting, $\mathbf{Z}_n^{\text{Lat}}$ can be succinctly written as $\sqrt{\bar{\alpha}_n}\mathbf{Z}_0^{\text{Lat}} + \sqrt{1-\bar{\alpha}_n}\mathbf{E}^n$, where $\mathbf{E}^n := [\epsilon_1^n, \epsilon_2^n, \ldots, \epsilon_F^n] \in \mathbb{R}^{T \times F}$ with $\epsilon_j^n \sim \mathcal{N}(0, \mathcal{I}_{T \times T})$. Finally, the ELBO loss we aim to minimize is:

$$\mathcal{L}_{\text{diff}} := \mathbb{E}_{n,\mathbf{E}^n}\left[\|\epsilon_\theta(\sqrt{\bar{\alpha}_n}\mathbf{Z}_0^{\text{Lat}} + \sqrt{1-\bar{\alpha}_n}\mathbf{E}^n, n, \mathbf{t}, \mathbf{ts}) - \mathbf{E}^n\|_2^2\right]. \tag{5}$$

The neural network $\epsilon_\theta$ predicts the error matrix $\mathbf{E}^n$ added in every diffusion step $n \sim \text{Unif}(\{1, 2, \ldots, N\})$. It takes noisy matrix $\mathbf{Z}_n^{\text{Lat}}$, normalized time-stamps $\mathbf{t} := \{t_1, t_2, \ldots, t_T\} = \{\frac{i}{T}\}_{i=1}^T$, diffusion step $n$, and the original time-stamps $\mathbf{ts}$ in the tabular dataset as inputs.

**Design of $\epsilon_\theta$:** The architecture of $\epsilon_\theta$ is given in Fig. 3. Diffusion step $n$ and a set of normalized time points $\mathbf{t}$ are encoded through positional encoding (in short PE) introduced in Vaswani et al. (2017). PE of $n$ lets the diffusion model know at which diffusion step the noisy matrix is, and PE of $\mathbf{t}$ encodes the sequential order of rows in the input matrix. But normalized time stamps provide only limited information on the orders of rows, and we find incorporating the encodings of timestamps in date-time format (i.e., *YEAR-MONTH-DATE-HOURS*), which can be commonly found in time series tabular data, significantly helps the diffusion training process. (See Table 5 in subsection L.) Cyclic encodings with sine and cosine functions are used for converting the date-time data to dense vectors: specifically, for $\mathbf{x} \in \{\text{YEAR}, \text{MONTH}, \text{DATE}, \text{HOURS}\}$ with Period $\in \{\text{total number of years in the dataset}, 12, 365, 24\}$, the conversion we used is as :

$$\left(\sin\left(\mathbf{x}/(\text{Period} \times 2\pi)\right), \cos\left(\mathbf{x}/(\text{Period} \times 2\pi)\right)\right). \tag{6}$$

Through equation 6, cyclic encodings give 8-dimensional unique representations of timestamps of the observed data in the table, and the encoded vector is fed to an MLP block equation 2 to match the dimension with those of the other inputs' encodings. The concatenated encodings of $(Z_n^{\text{Lat}}, \mathbf{n}, \mathbf{t}, \mathbf{ts})$ are fed into an MLP block, which gives a tensor $\mathbf{N} = [N_1, N_2, \ldots, N_T]^T \in \mathbb{R}^{T \times F}$. Inspired from Tian et al. (2023), Bi-directional RNN (Bi-RNN) (Schuster & Paliwal, 1997) is employed and $\mathbf{N}$ is fed to Bi-RNN as in Figure 3. After the applications of layer-normalization and FC layer, the network $\epsilon_\theta$ outputs $[\epsilon_\theta(t_1), \ldots, \epsilon_\theta(t_T)]^T \in \mathbb{R}^{T \times F}$ to estimate $\mathbb{E}^n$. The sampling process of the new latent matrix is deferred to Appendix H.

## 3 APPLICATION OF TIMEAUTODIFF: CONDITIONAL GENERATION ON TIME-VARYING SEQUENTIAL METADATA

In this section, we introduce C-TimeAutoDiff ('C' for conditional), where the model can generate heterogeneous outputs conditionally on time varying metatdata. Same with unconditional generation, the model consists of two learning stages on VAE and diffusion model. But unlike TimeAutoDiff,

VAE only needs to be trained on the output variables $\mathbf{x}$, as we need the trained decoder for the generation only. The metadata $\mathbf{c}$ can be conditioned on diffusion model directly without going through encoder layers. With a slight abuse of notation, let $\mathbf{Z}_0^{\mathrm{Lat}}$ be the latent matrix of the $\mathbf{x}$ the output of conditional generation, and we model $p(\mathbf{Z}_0^{\mathrm{Lat}}|\mathbf{c})$ through the diffusion model.

**Metadata Conditioning:** Instead of directly conditioning $\mathbf{c} = (c_{\mathrm{disc}}, c_{\mathrm{cont}})$ to the network architecture $\epsilon_\theta$, a preprocessing module on the condition is devised. Discrete metadata, $c_{\mathrm{disc}}$, is encoded through look-up table, and $c_{\mathrm{cont}}$ is processed trough an MLP block. Another MLP block is applied on the combined encoded metadata to learn the correlations among the discrete and continuous metadata features. For learning the temporal dependences, Bi-RNN and FC-layer are employed, where FC-layer is used to match the dimension of the encodings of $\mathbf{Z}_n^{\mathrm{Lat}}$. Visualizations of the network architecture $\epsilon_\theta$ of `C-TimeAutoDiff` are provided in the Appendix H.

## 4 NUMERICAL EXPERIMENTS

### 4.1 EXPERIMENTAL SETUP

**Datasets:** We select eight real-world time series tabular datasets consisting of both numerical and categorical features: Traffic, Pollution, Hurricane, AirQuality, ETTh1, Energy (single-sequence), and nasdaq100, card fraud (multi-sequence: sequences from multiple entities in one table). We provide the overall statistics and descriptions of these datasets in the Appendix C.

**Baselines:** To assess the quality of unconditionally generated time series data, we use 5 baseline models: (1) GAN based methods: TimeGAN (Yoon et al., 2019), DoppelGANger (Lin et al., 2020). (2) Diffusion based methods: Diffusion-TS (Yuan & Qiao, 2023), TSGM (Lim et al., 2023), (3) Parametric model: CPAR (Zhang et al., 2022).

**Evaluation Methods:** For the comprehensive quantitative evaluation of the synthesized data, we mainly focus on four criteria: (1) **Low-order statistic-** pair-wise column correlations and row-wise temporal dependences in the table are evaluated via *feature correlation score* (Kotelnikov et al., 2022) and *temporal discriminative score* (devised by us), respectively. (2) **High-order statistic-** the overall fidelities of the synthetic data in terms of joint distributional modeling are measured through *discriminative score* (Yoon et al., 2019). (3) The effectiveness of the synthetic data for **downstream** tasks is assessed through the predictive score (Yoon et al., 2019), where a predictive model (i.e., regressor or classifier) is trained using synthesized data and tested on real data (Mogren, 2016). (4) **Sampling times** (in sec.) are compared with other base-line methods. Detailed explanations for each metric are deferred in the Appendix G. Additionally, **generalizability** of the model is evaluated under "Distance to the Closest Record" (DCR; Park et al. (2018)) metric to ensure it draws samples from the distribution rather than memorizing the training data points (Appendix I). To evaluate the conditionally generated samples $\mathbf{x}^{\mathrm{con\text{-}syn}} \sim \mathbb{P}(\mathbf{x} \mid \mathbf{c})$, we employ the above metrics on the two datasets: $\mathcal{D}_{x,c}^{\mathrm{real}} := \{(\mathbf{x}^{\mathrm{real}}, \mathbf{c})\}$ and $\mathcal{D}_{x,c}^{\mathrm{synth}} := \{(\mathbf{x}^{\mathrm{con\text{-}syn}}, \mathbf{c})\}$ with $\mathbf{c}$ being fixed. As $\mathbb{P}(\mathbf{x}, \mathbf{c}) = \mathbb{P}(\mathbf{c})\mathbb{P}(\mathbf{x} \mid \mathbf{c})$, in this way, we measure conditional relations of $\mathbf{x}^{\mathrm{con\text{-}syn}}$ and $\mathbf{c}$ as well as the fidelity of $\mathbf{x}^{\mathrm{con\text{-}syn}}$ to $\mathbf{x}^{\mathrm{real}}$.

**Parameter setting:** In Appendix K, we present the parameter settings of VAE and DDPM in our model. Unless otherwise specified, they are universally applied to the entire dataset in the experiments conducted in this paper. Additionally, we study how the sizes of network architectures in DDPM and VAE, training epochs for both models, and noise schedulers (linear vs quadratic) in DDPM affect the performances of the model.

### 4.2 FIDELITY AND UTILITY GUARANTEES OF SYNTHETIC DATA

**Unconditional Generation:** Table 1 shows that our `TimeAutoDiff` consistently outperforms other baseline models in terms of almost all metrics both for single- and multi-sequence generation tasks. It significantly improves the (temporal) discriminative and feature correlation scores in all datasets over the baseline models. `TimeAutoDiff` also dominates the predictive score metric. (We train a classifier to predict a column in the dataset to measure the predictive score. The columns predicted in each dataset are listed in Table 4 in Appendix C.) But for some datasets, the performance gaps with the second-best model are negligible, for instance, TSGM for Hurricane and AirQuality datasets. It is intriguing to note that the predictive scores can be good even when the data fidelity is low. The GAN-based models are faster in terms of sampling time compared to the diffusion-based

| Metric | Methods | Single-Sequence | | | | Multi-Sequence | |
|---|---|---|---|---|---|---|---|
| | | Traffic | Pollution | Hurricane | AirQuality | Card Transaction | nasdaq100 |
| Discriminative Score (The lower, the better) | TimeAutoDiff | **0.026(0.014)** | **0.016(0.009)** | **0.047(0.016)** | **0.061(0.013)** | **0.215(0.058)** | **0.067(0.046)** |
| | Diffusion-ts | 0.202(0.021) | 0.133(0.015) | 0.181(0.018) | 0.134(0.016) | N.A. | N.A. |
| | TSGM | 0.500(0.000) | 0.488(0.010) | 0.482(0.020) | 0.452(0.009) | N.A. | N.A. |
| | TimeGAN | 0.413(0.057) | 0.351(0.053) | 0.254(0.062) | 0.460(0.020) | 0.482(0.037) | 0.267(0.115) |
| | DoppelGANger | 0.258(0.215) | 0.100(0.103) | 0.176(0.099) | 0.211(0.116) | 0.485(0.025) | 0.071(0.032) |
| | CPAR | 0.498(0.002) | 0.500(0.000) | 0.500(0.000) | 0.499(0.001) | 0.500(0.000) | 0.143(0.120) |
| | Real vs Real | 0.053(0.009) | 0.048(0.017) | 0.048(0.017) | 0.040(0.011) | 0.225(0.094) | 0.190(0.051) |
| Predictive Score (The lower, the better) | TimeAutoDiff | **0.203(0.014)** | **0.008(0.000)** | **0.098(0.026)** | **0.005(0.001)** | **0.001(0.000)** | 10.863(0.716) |
| | Diffusion-ts | 0.231(0.007) | 0.013(0.000) | 0.306(0.076) | 0.017(0.002) | N.A. | N.A. |
| | TSGM | 0.247(0.002) | 0.009(0.000) | 0.290(0.007) | 0.006(0.000) | N.A. | N.A. |
| | TimeGAN | 0.297(0.008) | 0.043(0.000) | 0.180(0.027) | 0.057(0.011) | 0.130(0.022) | 9.597(0.016) |
| | DoppelGANger | 0.300(0.005) | 0.282(0.028) | 0.214(0.000) | 0.060(0.009) | 0.004(0.006) | 11.556(1.093) |
| | CPAR | 0.263(0.003) | 0.032(0.009) | 0.420(0.055) | 0.030(0.007) | 0.132(0.035) | **8.270(0.019)** |
| | Real vs Real | 0.206(0.012) | 0.010(0.000) | 0.098(0.026) | 0.005(0.001) | 0.001(0.000) | 9.281(0.009) |
| Temporal Discriminative Score (The lower, the better) | TimeAutoDiff | **0.047(0.018)** | **0.014(0.013)** | **0.026(0.024)** | **0.033(0.014)** | **0.290(0.040)** | 0.159(0.140) |
| | Diffusion-ts | 0.199(0.028) | 0.165(0.084) | 0.247(0.093) | 0.183(0.064) | N.A. | N.A. |
| | TSGM | 0.499(0.001) | 0.499(0.001) | 0.497(0.002) | 0.499(0.000) | N.A. | N.A. |
| | TimeGAN | 0.429(0.050) | 0.397(0.060) | 0.465(0.025) | 0.457(0.014) | 0.497(0.007) | 0.419(0.140) |
| | DoppelGANger | 0.400(0.039) | 0.444(0.050) | 0.464(0.028) | 0.335(0.091) | 0.362(0.097) | 0.497(0.000) |
| | CPAR | 0.436(0.073) | 0.492(0.021) | 0.497(0.009) | 0.493(0.010) | 0.470(0.041) | 0.404(0.099) |
| | Real vs Real | 0.061(0.011) | 0.044(0.009) | 0.039(0.012) | 0.050(0.017) | 0.360(0.051) | 0.150(0.090) |
| Feature Correlation Score (The lower, the better) | TimeAutoDiff | **0.022(0.014)** | **1.244(0.844)** | **0.074(0.013)** | **0.463(0.080)** | 0.078(0.137) | 0.243(0.012) |
| | Diffusion-ts | 2.148(1.439) | 1.716(1.096) | 1.881(1.208) | 0.716(0.141) | N.A. | N.A. |
| | TSGM | 2.092(1.485) | 1.710(0.705) | 0.424(0.249) | 0.543(0.077) | N.A. | N.A. |
| | TimeGAN | 1.243(0.535) | 2.068(1.093) | 2.151(1.113) | 0.865(0.123) | 2.301(0.723) | 1.488(1.069) |
| | DoppelGANger | 0.885(0.737) | 2.371(0.875) | 2.380(0.798) | 1.628(0.231) | 1.550(1.034) | 1.035(0.818) |
| | CPAR | 0.538(0.336) | 1.280(0.931) | 0.965(0.287) | 1.552(0.220) | 0.295(0.294) | 0.514(0.445) |
| | Real vs Real | 0.000(0.000) | 0.000(0.000) | 0.000(0.000) | 0.000(0.000) | 0.000(0.000) | 0.000(0.000) |
| Sampling Time (in Sec) (The lower, the better) | TimeAutoDiff | 3.512(0.065) | 3.947(0.070) | 3.740(0.132) | 3.945(0.103) | 3.384(0.064) | 3.133(0.129) |
| | Diffusion-ts | ≫ | ≫ | ≫ | ≫ | N.A. | N.A. |
| | TSGM | ≫ | ≫ | ≫ | ≫ | N.A. | N.A. |
| | TimeGAN | 0.127(0.056) | 0.113(0.058) | 0.125(0.060) | 0.131(0.060) | 0.051(0.051) | 0.047(0.039) |
| | DoppelGANger | **0.011(0.002)** | **0.014(0.001)** | **0.010(0.003)** | **0.017(0.003)** | **0.018(0.004)** | **0.041(0.001)** |
| | CPAR | 17.466(0.734) | 18.597(0.558) | 15.839(0.324) | 29.816(0.846) | 141.425(2.435) | 112.506(2.152) |

Table 1: The experimental results of single-sequence and multi-sequence time series tabular data generations under the Discriminative, Predictive, Temporal Discriminative, and Feature Correlation scores. Sampling times of each model over 6 datasets are recorded in seconds. The symbol ≫ denotes that the sampling time exceeds 300 seconds, and 'N.A.' means 'Not Applicable'. The bolded number indicates the best-performed result. For each metric, the mean and standard deviation (in parenthesis) of 10 scores from one generated synthetic data are recorded in the table. For recording the sampling time, 10 synthetic data are generated from the trained diffusion model. The 'Real Data' serves as a baseline, where each metric is computed under Real vs Real.

| Metric | Methods | Single-Sequence | | | | | |
|---|---|---|---|---|---|---|---|
| | | Traffic | Pollution | Hurricane | AirQuality | ETTh1 | Energy |
| Discriminative Score | C-TimeAutoDiff | **0.078(0.038)** | **0.056(0.017)** | **0.014(0.005)** | 0.090(0.007) | **0.036(0.008)** | **0.113(0.070)** |
| | Real vs Real | 0.091(0.021) | 0.067(0.020) | 0.081(0.009) | **0.085(0.027)** | 0.051(0.011) | 0.270(0.028) |
| Predictive Score | C-TimeAutoDiff | 0.113(0.007) | **0.008(0.000)** | **0.004(0.000)** | 0.060(0.009) | **0.048(0.002)** | **0.228(0.005)** |
| | Real vs Real | **0.107(0.001)** | 0.008(0.000) | **0.058(0.010)** | 0.004(0.000) | 0.051(0.001) | 0.230(0.003) |
| Temporal Discriminative Score | C-TimeAutoDiff | **0.123(0.034)** | **0.081(0.027)** | **0.048(0.025)** | **0.116(0.018)** | **0.045(0.015)** | **0.224(0.013)** |
| | Real vs Real | 0.134(0.015) | 0.083(0.019) | 0.072(0.019) | 0.138(0.014) | 0.074(0.014) | 0.300(0.031) |
| Feature Correlation Score | C-TimeAutoDiff | **0.012(0.003)** | **0.026(0.008)** | **0.175(0.032)** | **0.011(0.002)** | **0.014(0.002)** | **0.029(0.007)** |
| | Real vs Real | 0.000(0.000) | 0.000(0.000) | 0.000(0.000) | 0.000(0.000) | 0.000(0.000) | 0.000(0.000) |

Table 2: Time varying metadata conditional generations: the experiments conducted over 6 single-sequence datasets with sequence length set as $T = 96$. See the caption of Figure 14 (Appendix N) for output and condition pairs for each dataset used for the experiments. Overall, C-TimeAutoDiff performs well, achieving results comparable to the Real vs Real baseline over the test dataset.

models. These results are expected, as diffusion-based models require multiple denoising steps for sampling, whereas GAN-based models generate samples in a single step. Among diffusion-based models, our model shows the best performance for sampling time. In the Appendix O and J, we provide additional experiments on more metrics such as volatility, moving averages, Maximum Mean Discrepancy (MMD) and entropy for diversity (Nikitin et al., 2023).

**Conditional Generation:** To test if C-TimeAutoDiff generalizes to unseen conditions, we randomly split the dataset into train/test (80%/20%) sets. Synthetic data is generated with the same size as the test dataset. The sequence length is set as 96. Qualities of the data are evaluated under the introduced metrics in Table 5. To the best of our knowledge, there are no existing baseline methods that perform similar tasks as C-TimeAutoDiff. Instead, metrics computed over Real vs Real are used as the baseline. Our model performs on-par or even better than the Real vs Real baseline.

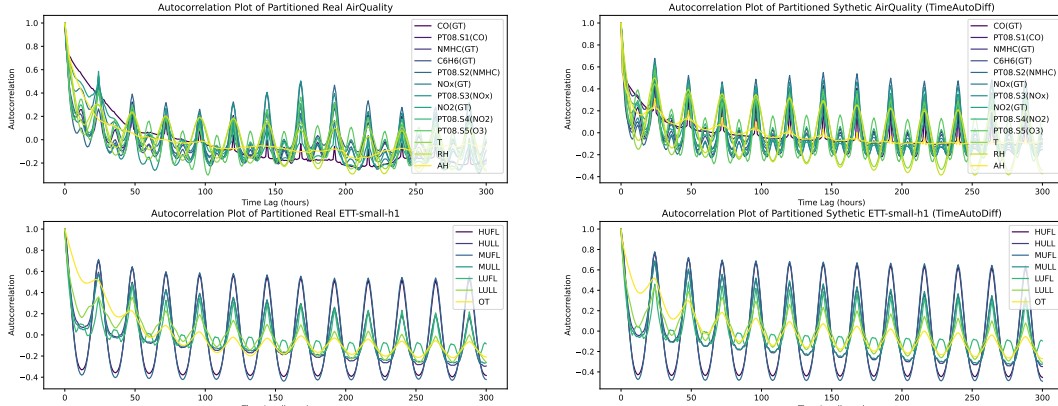

Figure 5: Real (left) vs. Synthetic (right): Autocorrelation plots with a time lag of 300 (hours) for the AirQuality (Top) and the ETTh1 (Bottom) datasets. Sequence length is set as $T = 500$.

**Temporal & Feature Dependences:** Aside from quantitative evaluations under the mentioned metrics, as illustrated in Fig. 5, autocorrelation plots for both real (AirQuality and ETTh1) and synthetic data reveal that the `TimeAutoDiff` successfully captures the complex and long temporal correlations of sequences, i.e., $T = 500$. Additionally, the similar patterns observed for each feature in the real and synthetic data demonstrate the model's ability to capture the correlations along the feature dimension. We provide more visualizations across various datasets in the Appendix M.

Additional experiments on ablation, scalability and the adaptive choices on $\beta$ in VAE are deferred in the Appendix L.

### 4.3 Time-Varying Metadata Conditional Generation

We further provide numerical validations that `C-TimeAutoDiff` indeed learn the conditional distribution $\mathbb{P}(\mathbf{x}|\mathbf{c})$ of both $\mathbb{P}(\text{'Cont Var.'}|\text{'Disc Var.'})$ and $\mathbb{P}(\text{'Disc Var.'}|\text{'Cont Var.'})$ under synthetic data setting. Additionally, we explore its application in counterfactual scenario analysis with real-world Traffic data, investigating how weather sequences affect traffic volume.

**Synthetic Setting:** Real-world data often involves complex correlations and confounding factors, making it difficult to establish strict causal relationships. To validate that `C-TimeAutoDiff` can effectively learn conditional rules, we use a synthetic dataset with variables 'Temperature' and 'Weather'. The 'Temperature' is generated over 10,000 time points as:

$$\text{Temp}(t) = 15 + 10 \sin\left(2\pi t/365\right) + \mathcal{N}(0, 2^2),$$

where Temp(t) follows a sinusoidal pattern with added Gaussian noise. Based on the generated 'Temperature', the categorical 'Weather' variable is derived as follows: 'Sunny' if Temp $> 20$, 'Cloudy' if $10 < \text{Temp} \le 20$, and 'Rainy' if $0 < \text{Temp} \le 10$. We set the time window as $T = 48$ (hours) and train the model to learn two conditional distributions: $\mathbb{P}(\text{Temp}|\text{Weather})$, which predicts temperature given weather, and $\mathbb{P}(\text{Weather}|\text{Temp})$, which predicts weather given temperature.

Fig 6 (top 3) demonstrates the model's ability to generate 'Temperature' sequences corresponding to specific weather conditions under three scenarios: (1) constant weather conditions over three consecutive 48-time periods ('Sunny', 'Cloudy', 'Rainy'), (2) a repeating pattern of weather labels (e.g., 16 'Rainy', 16 'Cloudy', 16 'Sunny'), and (3) random alternating patterns of 'Cloudy' and 'Rainy'. The results show distinct separations in the temperature sequences generated for each weather condition, validating the model's ability to learn $\mathbb{P}(\text{Cont Var.}|\text{Disc Var.})$. Similarly, Fig 6 (bottom 3) demonstrates the reverse case. When conditioned on 'Temperature' values generated in the previous scenarios, the model correctly predicts the corresponding 'Weather' labels at each time step, further validating its ability to learn $\mathbb{P}(\text{Disc Var.}|\text{Cont Var.})$.

**Traffic Data:** To evaluate `C-TimeAutoDiff` on real-world data, we use the Traffic dataset with 'Traffic Volume' (continuous) as the output and 'Weather-main' (categorical) as the conditional variable. 'Weather-main' includes labels such as {'Clear', 'Rain', 'Squall', 'Cloudy'}, among others. Intuitively, we expect lower traffic volumes during adverse weather conditions (e.g., 'Squall', 'Rain')

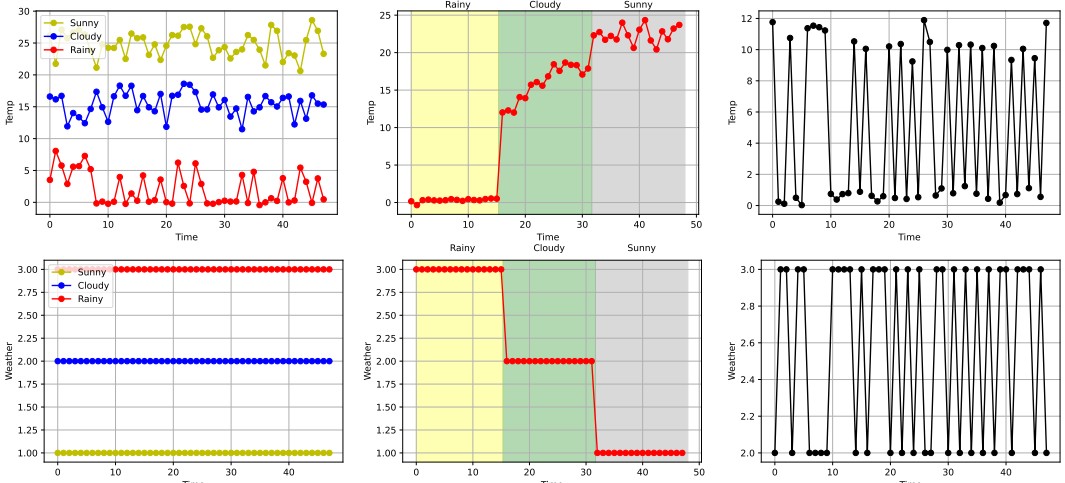

Figure 6: Empirical validations of the model's ability to learn the conditional probability distributions $\mathbb{P}\big(\text{'Temperature'}|\text{'Weather'}\big)$ (top 3 panels) and $\mathbb{P}\big(\text{'Weather'}|\text{'Temperature'}\big)$ (bottom 3 panels).

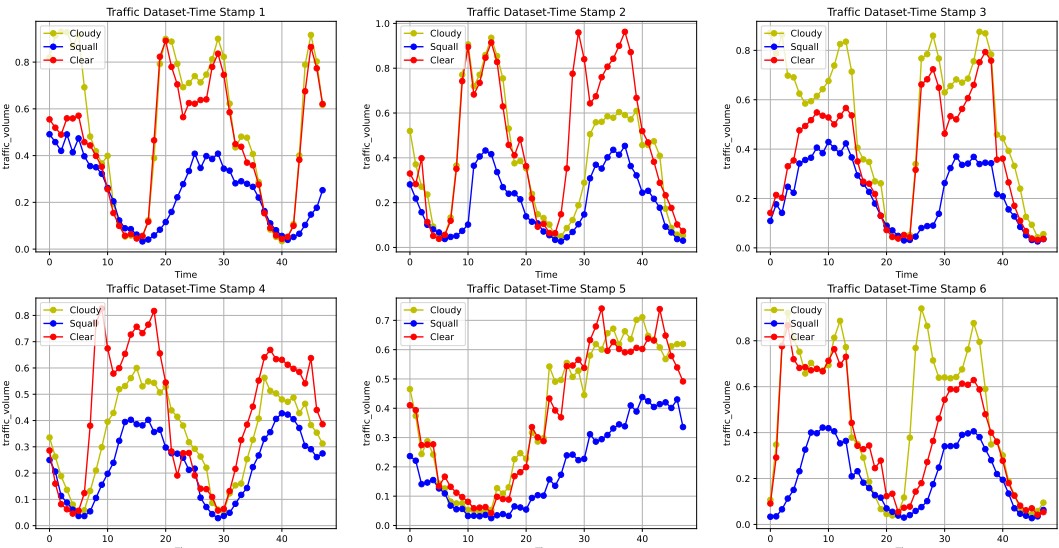

Figure 7: We choose arbitary 6 timestamp sequences in dataset, and give the models labels of ['Cloudy', 'Squall', 'Clear'] weather-conditions. The traffic-volume axis is normalized.

and higher traffic volumes during good weather (e.g., 'Clear', 'Cloudy'). We test the model under three weather scenarios: 'Cloudy', 'Squall', and 'Clear', using six different timestamp sequences to observe patterns. As shown in the results, 'Traffic Volume' is consistently lower during 'Squall' compared to 'Cloudy' and 'Clear', while no significant differences are observed between 'Clear' and 'Cloudy'. These findings confirm the model's ability to reflect expected traffic patterns under different weather conditions.

## 5 DISCUSSIONS

This paper introduces `TimeAutoDiff`, a novel time series tabular data synthesizer designed for multi-dimensional, heterogeneous features. Leveraging a latent diffusion model with a specialized VAE, it achieves high fidelity and utility guarantees. The model supports time-varying metadata conditional generation, enabling applications across scientific and engineering domains. It also lays the groundwork for tasks such as *missing data imputation*, *privacy guarantees*, *interpretability*, and *extension to foundational models*, all of which rely on precise modeling of $\mathbb{P}(\mathbf{x}_T, \mathbf{x}_{T-1}, \dots, \mathbf{x}_1)$. Further discussions are provided in Appendix A.

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

## A  DISCUSSIONS ON FUTURE TOPICS WITH RELEVANT LITERATURE

In this subsection, we further discuss about the four possible extensions of `TimeAutoDiff` in sequel: (1) Missing data imputation; (2) Privacy guarantees; (3) Interpretability of generated time series data; (4) Extension to foundational model.

**(1) Missing data imputation** is an important application of tabular data synthesis. In the literature, CSDI (Tashiro et al., 2021) study the imputations of continuous time series tabular data through diffusion-based framework. The main idea is to employ the specially designed masks; masking the observed data, and to let the model predict the masked values in the observations, i.e., self-supervised learning. Then, the trained model can impute the real missing parts of the table by thinking them as masked observations. We conjecture the similar idea can be easily adopted in the framework of `TimeAutoDiff`. In i.i.d. row setting (each row from the same distribution), several papers (Zhang et al., 2023a; 2024) study the imputation problem of tabular data with heterogeneous features through diffusion-based synthesizers. Zhang et al. (2023a) directly used the pre-trained unconditional latent diffusion model, analogous to inpainting tasks of images, for the imputation. Zhang et al. (2024) employed the concept of EM-algorithm. Specifically, the former work, Zhang et al. (2023a), utilized the fact that the transformer maps the input data to latent space deterministically, where transformer is used for the main backbone architecture in VAE.

**(2) Privacy Guarantees** is one of the main motivations of synthetic data. Specifically, in the time series domain, data from the healthcare and financial sectors is ubiquitous, but it often comes with significant privacy concerns. We hope the synthetic data does not leak any private information of the original data, while preserving good fidelities. `TimeAutoDiff` lays the foundation for guaranteeing such privacy concerns with the generated synthetic data. In the vision domain, differential privacy guarantees (Dwork, 2006) of synthetic images from diffusion-based models have been investigated by several researchers (Dockhorn et al., 2022; Ghalebikesabi et al., 2023; Lyu et al., 2023). Specifically, Lyu et al. (2023) studied DP-guarantees of latent diffusion model by fine-tuning the attention module of noise predictor in their diffusion model, and claim their synthetic images both have good fidelities and DP-guarantees.

Nonetheless, it is still not clear how the same idea can be applied to time series synthetic data (or regular tabular data), as differentially private time series data is frequently challenging to interpret (Yoon et al., 2020). In this regard, another privacy criterion, $\varepsilon$-identifiability (Yoon et al., 2020) (with $\varepsilon \in [0, 1]$) can be considered as another alternative. The distance between synthetic and original data is measured through Euclidean distance, and we want at least $(1 - \varepsilon)$-proportion of the synthetic data to be distinguishable (or different enough) from the original data. Under this criterion, we conjecture `TimeAutoDiff` can be extended to the synthesizer with a (theoretically-provable) privacy guarantees. The idea can be underpinned around several recent results on diffusion model (Zhang et al., 2023b; Bodin et al., 2024). Zhang et al. (2023b) showed that there exist closed-form solutions of noise predictors for every diffusion step of noisy training data points. This means that we can trace back the latent vectors (or matrix) where the original training data points are generated from. Recent findings (Bodin et al., 2024) suggest that a proper linear combination of data in the latent space can produce a new semantically meaningful dataset in the original space. Combining the fact that the mapping from the latent space to the original space is Lipschitz continuous (Zhang et al., 2023b) through deterministic sampling (probability-flow), we might be able to have controls over the generations of time series synthetic data, whose Euclidean distances from training data points are away from the training data points. This idea is naturally related to the diversity of generated data as well.

**(3) Interpretability** of the generated time series data is another crucial aspect that time series synthesizer should possess. In many practical applications, for instance, in financial sector, stakeholders and domain experts may be hesitant to rely on synthesis models that are difficult to interpret, as they need to understand and trust the model's behavior, especially when dealing with critical or high-risk scenarios. The current version of `TimeAutoDiff` does not have the luxury of generating interpretable results, but this can be easily adopted by following the previous works. Specifically, we want to point out readers TimeVAE (Desai et al., 2021) and Diffusion-TS (Yuan & Qiao, 2023), which both focus on building a synthesizer with interpretability. Specifically, TimeVAE adopted a sophisticatedly designed decoder in VAE, which has *trend*, *seasonality*, and *residual*

blocks for signal decompositions. Similarly, Diffusion-TS also design a sophisticated decoder for the decomposition of signals into trend, seasonality, and residual, where they employ the latent diffusion framework. Both of these ideas can be directly employed in `TimeAutoDiff`, where the current decoder is set as an MLP block for simplicity.

**(4) Extension to foundational model** is another promising route the `TimeAutoDiff` can take. Recently, we have been seeing a wave of foundational models research on time series domain (Cao et al., 2024; Liu et al., 2024; Das et al., 2023; Yang et al., 2024a). These models can accommodate multiple tables from cross domains, enabling multiple time series tasks in one model; for instance, forecasting, anomaly detection, imputation, and synthetic data generation (See Cao et al. (2024).) Among them, Cao et al. (2024) devised cleverly designed masks, which provide the unifying framework to do the four abovementioned tasks under diffusion-based framework. Nonetheless, their methods are confined to the continuous data modality, and not clear how the model can be extended to heterogeneous features, leaving the great future opportunities for TimeAutoDiff to be extended. We also conjecture the synthetic data from `TimeAutoDiff` can be beneficial to improving quality of forecasting foundation model i.e., see Section 5 in (Das et al., 2023).

**(5) Bias from conditional metadata generation:** Generated data can indeed be biased with respect to conditional metadata, arising from various factors. Bias in the training data, such as inherent associations between metadata and outputs, may lead the model to replicate these biases, for instance, generating disproportionately high traffic volumes for "Clear" weather even when the true relationship is less deterministic. Imbalanced metadata distributions further exacerbate this issue, as underrepresented conditions in the training set often result in less reliable outputs for those conditions, such as biased outcomes for minority demographic groups in healthcare datasets. Simplified assumptions in the model, such as assuming linear relationships between metadata and outputs, can overlook complex dependencies, producing data that fails to reflect the true conditional distribution. Noise injection, a feature of models like diffusion models and VAEs, can introduce additional bias if the noise interacts with metadata in unexpected ways, particularly for rare metadata values. Furthermore, limitations in conditional architectures, such as inadequate metadata encoding, can prevent the model from capturing nuanced dependencies, leading to misaligned outputs. To mitigate such biases, ensuring balanced training data, employing robust metadata encoding techniques, applying regularization or fairness constraints, performing post-generation bias audits, and designing disentangled latent spaces are crucial steps. While conditional generative models aim to align generated data with metadata, addressing these biases is essential to ensure fairness and reliability.

## B  COMPUTING RESOURCES

We ran the main model on a computer equipped with an Intel(R) Core(TM) i9-14900KF 3.20 GHz, an NVIDIA GeForce RTX 4090 with 24GB VRAM.

## C  DATASETS AND DATA PROCESSING STEPS

We used six single-sequence and two multi-sequence time-series datasets for our experiments. The statistical information of datasets used in our experiments is in Table 3.

**Single-sequence:** We select the first 2000 rows from each single sequence dataset for our experiments. We split our data into windows of size $T$, leading us to have the tensor of size $(2000 - T + 1) \times T \times F$. (We truncate the rows of the tables because of the memory issues we encounter for large $T$ (e.g., $T = 900$).) Recall the $F$ denote the number of features in the table.

- **Traffic** (UCI) is a single-sequence, mixed-type time-series dataset describing the hourly Minneapolis-St Paul, MN traffic volume for Westbound I-94. The dataset includes weather features and holidays for evaluating their impacts on traffic volume. (URL: `https://archive.ics.uci.edu/dataset/492/metro+interstate+traffic+volume`)
- **Pollution** (UCI) is a single-sequence, mixed-type time-series dataset containing the PM2.5 data in Beijing between Jan 1st, 2010 to Dec 31st, 2014. (URL: `https://archive.ics.uci.edu/dataset/381/beijing+pm2+5+data`)

| Dataset | # of Rows | #-Cont. | #-Disc. | Seq. Type | Pred Score Col. |
|---------|-----------|---------|---------|-----------|-----------------|
| Traffic | 48205 | 3 | 5 | Single | `traffic volume` |
| Pollution | 43825 | 5 | 3 | Single | `lr` |
| Hurricane | 9937 | 4 | 4 | Single | `seasonal` |
| AirQuality | 9358 | 1 | 12 | Single | `AH` |
| ETTh1 | 17431 | 7 | 0 | Single | `OT` |
| Energy | 19736 | 27 | 1 | Single | `rv2` |
| Card Transaction | 20000 | 2 | 6 | Multi | `Is Fraud?` |
| nasdaq100 | 18231 | 3 | 4 | Multi | `Industry` |

Table 3: Datasets used for our experiments. The date time column is considered as neither continuous nor categorical. The 'Seq.Type' denotes the time series data type: single- or multi-sequence data. The 'Pred Score Col' denotes columns in each dataset used for measuring predictive scores.

- **Hurricane** (NHC) is a single sequence, mixed-type time-series dataset of the monthly sales revenue (2003-2020) for the tourism industry for all 67 counties of Florida which are prone to annual hurricanes. This dataset is used as a spatio-temporal benchmark dataset for forecasting extreme events and anomalies (Farhangi et al., 2023). (URL: `https://www.nhc.noaa.gov/data/`)

- **AirQuality** (UCI) is a single sequence, mixed-type time-series dataset containing the hourly averaged responses from a gas multisensor device deployed on the field in an Italian city. (URL: `https://archive.ics.uci.edu/dataset/360/air+quality`)

- **ETTh1** (Github: Zhou et al. (2021)) is a single sequence, continuous only time-series dataset, recording hourly level ETT (i.e., Electricity Transformer Temperature), which is a crucial indicator in the electric power long-term deployment. Specifically, the dataset combines short-term and long-term periodical patterns, long-term trends, and many irregular patterns. (URL: `https://github.com/zhouhaoyi/ETDataset/tree/main`)

- **Energy** (Kaggle) is a single sequence time-series dataset. The dataset, spanning 4.5 months, includes 10-minute interval data on house temperature and humidity via a ZigBee sensor network, energy data from m-bus meters, and weather data from Chievres Airport, Belgium, with two random variables added for regression model testing. (URL: `https://www.kaggle.com/code/gaganmaahi224/appliances-energy-time-series-analysis`)

**Multi-sequence:** The sequences in the multi-sequence data vary in length from one entity to another, so we selected entities with sequences longer than $T = 200$ and $T = 177$ and truncated them to a uniform length of $T$ for the "card transaction" and "nasdaq100" datasets.

- **Card Transaction** is a multi-sequence, synthetic mixed-type time-series dataset created by Padhi et al. (2021a) using a rule-based generator to simulate real-world credit card transactions. We selected 100 users (i.e., entities) for our experiment. In the dataset, we choose {*'Card', 'Amount', 'Use Chip', 'Merchant', 'MCC', 'Errors?', 'Is Fraud?'*} as features for the experiment. (URL: `https://github.com/IBM/TabFormer/tree/main`)

- **nasdaq100** is a multi-sequence, mixed-type time-series dataset consisting of stock prices of 103 corporations (i.e., entities) under nasdaq 100 and the index value of nasdaq 100. This data covers the period from July 26, 2016 to April 28, 2017, in total 191 days. (URL: `https://cseweb.ucsd.edu/~yaq007/NASDAQ100_stock_data.html`)

# D  PRE- AND POST-PROCESSING STEPS IN TIMEAUTODIFF

It is essential to pre-process the real tabular data in a form that the machine learning model can extract the desired information from the data properly. We divide the heterogeneous features into two categories; (1) continuous, and (2) discrete. Following is how we categorize the variables and process each feature type. Let x be the column of a table to be processed.

1. ***Continuous feature***: If $\mathbf{x}$'s entries are real-valued continuous, we categorize $\mathbf{x}$ as a numerical feature. Moreover, if the entries are integers with more than 25 distinct values (e.g., "Age"), then $\mathbf{x}$ is categorized as a continuous variable. Here, 25 is a user-specified threshold. We employ min-max scaler (Yoon et al., 2019) to ensure the pre-processed numerical features are within the range of $[0, 1]$. Hereafter, we denote $\mathbf{x}_{\text{Num}}^{\text{Proc}}$ as the processed column.

2. ***Discrete / Categorical feature***: If $\mathbf{x}$'s entries have string datatype, we categorize $\mathbf{x}$ as a discrete feature (e.g., "Gender"). Additionally, the $\mathbf{x}$ with less than 25 distinct integers is categorized as a discrete feature. For pre-processing, we simply map the entries of $\mathbf{x}$ to the integers greater than or equal to $0$, and further divide the data type into two parts; binary and categorical, denoting them as $\mathbf{x}_{\text{Bin}}^{\text{Proc}}$ and $\mathbf{x}_{\text{Cat}}^{\text{Proc}}$. Here, $\mathbf{x}_{\text{Cat}}^{\text{Proc}}$ denotes the discrete variables with more than 3 labels or categories.

3. ***Post-processing step***: After the `TimeAutoDiff` model generates a synthetic dataset, it must be restored to its original format. For continuous features, this is achieved through inverse transformations, (i.e., reversing min-max scaling). Integer labels in discrete features are mapped back to their original categorical or string values.

## E    COMPARISON TABLE OF TIMEAUTODIFF WITH CURRENT LITERATURE

Table E compares TimeAutoDiff with other time series synthesizers in the literature under seven different aspects. Additionally, we provide further detailed comparisons between our model and Diffusion-TS (Yuan & Qiao, 2023) / TimeDiff (Tian et al., 2023). Diffusion-TS's main purpose is to generate time series data with interpretability. They employ the Autoencoder + DDPM framework, employing transformers as encoder and decoder for obtaining the disentangled representations of time series. The main difference between Diffusion-TS and ours is on the problem setting that their assumption on the signal is only restricted to continuous time series, whereas ours is focused on the heterogeneous features. Diffusion-TS lies on the assumption that the signal is decomposable into three main parts: trend, seasonality, and noise. However, the decomposition of heterogeneous features, specifically discrete variables is not well defined in the literature, it is beyond the scope of our work, requiring further research. TimeDiff integrates two types of diffusion models to handle heterogeneous features in EHR datasets, employing DDPM for continuous variables and multinomial diffusion (Hoogeboom et al., 2021) for discrete variables. In contrast, our approach leverages a VAE to project time series data into a latent space and utilizes DDPM exclusively for modeling the time series within this latent representation, which is continuous.

| Models | Hetero. | Single-Seq. | Multi-Seq. | Cond. Gen. | Applicability | Code | Sampling Time |
|---|---|---|---|---|---|---|---|
| `TimeAutoDiff` | ✓ | ✓ | ✓ | ✓ | ✓ | ✓ | 3 |
| TimeDiff (Tian et al., 2023) | ✓ | ✓ | ✗ | ✗ | ✗ | ✗ | – |
| Diffusion-ts (Yuan & Qiao, 2023) | ✗ | ✓ | ✗ | ✗ | ✓ | ✓ | 5 |
| TSGM (Lim et al., 2023) | ✗ | ✓ | ✗ | ✗ | ✓ | ✓ | 6 |
| TimeGAN (Yoon et al., 2019) | ✗ | ✓ | ✗ | ✗ | ✓ | ✓ | 2 |
| DoppelGANger (Lin et al., 2020) | ✗ | ✓ | ✗ | ✗ | ✓ | ✓ | 1 |
| EHR-M-GAN (Li et al., 2023) | ✓ | ✓ | ✗ | ✗ | ✗ | ✓ | – |
| CPAR (Zhang et al., 2022) | ✓ | ✗ | ✓ | ✗ | ✓ | ✓ | 4 |
| TabGPT (Padhi et al., 2021b) | ✓ | ✗ | ✓ | ✗ | ✗ | ✓ | – |

Table 4: A comparison table that summarizes `TimeAutoDiff` against baseline methods, evaluating metrics like heterogeneity, single- and multi-sequence data generation, conditional generation, applicability (i.e., whether the model is not designed for specific domains), code availability, and sampling time. Baseline models without domain specificity and with available code are used for numerical comparisons. The sampling time column ranks models by their speed, with lower numbers indicating faster sampling.

## F    DENOISING DIFFUSION PROBABILISTIC MODEL

Ho et al. (2020) proposes the denoising diffusion probabilistic model (DDPM) which gradually adds *fixed* Gaussian noise to the observed data point $\mathbf{x}_0$ via known variance scales $\beta_n \in (0, 1)$, $n \in \{1, \ldots, N\}$ at the diffusion step $n$. This process is referred as *forward process* in the diffusion

model, perturbing the data point and defining a sequence of noisy data $\mathbf{x}_1, \mathbf{x}_2, \ldots, \mathbf{x}_N$:

$$q(\mathbf{x}_n \mid \mathbf{x}_{n-1}) = \mathcal{N}(\mathbf{x}_n; \sqrt{1-\beta_n}\mathbf{x}_{n-1}, \beta_n\mathcal{I}), \quad q(\mathbf{x}_{1:N} \mid \mathbf{x}_0) := \prod_{n=1}^{N} q(\mathbf{x}_n \mid \mathbf{x}_{n-1}).$$

Since the transition kernel is Gaussian, the conditional probability of the $\mathbf{x}_n$ given its original observation $\mathbf{x}_0$ can be succinctly written as:

$$q(\mathbf{x}_n \mid \mathbf{x}_0) = \mathcal{N}\big(\mathbf{x}_n \mid \sqrt{\bar{\alpha}_n}\mathbf{x}_0, (1-\bar{\alpha}_n)\mathcal{I}\big),$$

where $\alpha_n = 1 - \beta_n$ and $\bar{\alpha}_n = \Pi_{k=1}^{n}\alpha_k$. Setting $\beta_n$ to be an increasing sequence, for large enough $N$, leads $\mathbf{x}_N$ to the isotropic Gaussian.

Training objective of DDPM is to maximize the evidence lower bound (in short ELBO) of the log-likelihood $\mathbb{E}_{\mathbf{x}_0}[\log p_\theta(\mathbf{x}_0)]$ as follows;

$$\mathbb{E}_q\bigg[\log p_\theta(\mathbf{x}_0 \mid \mathbf{x}_1) - \mathcal{D}_{\mathbf{KL}}(q(\mathbf{x}_N \mid \mathbf{x}_0) \,\|\, p(\mathbf{x}_N)) - \sum_{n=1}^{N} \mathcal{D}_{\mathbf{KL}}(q(\mathbf{x}_{n-1} \mid \mathbf{x}_n, \mathbf{x}_0) \,\|\, p_\theta(\mathbf{x}_{n-1} \mid \mathbf{x}_n))\bigg].$$

The first two terms in the expectation are constants, and the third KL-divergence term needs to be controlled. Interestingly, the conditional probability $q(\mathbf{x}_{n-1} \mid \mathbf{x}_n, \mathbf{x}_0)$ can be driven in the closed-form solution:

$$q(\mathbf{x}_{n-1} \mid \mathbf{x}_n, \mathbf{x}_0) = \mathcal{N}\left(\mathbf{x}_{n-1} \mid \frac{\sqrt{\bar{\alpha}_{n-1}}\beta_n}{1-\bar{\alpha}_n}\mathbf{x}_0 + \frac{\sqrt{\alpha_n}(1-\bar{\alpha}_{n-1})}{1-\bar{\alpha}_n}\mathbf{x}_n, \frac{1-\bar{\alpha}_{n-1}}{1-\bar{\alpha}_n}\beta_n\mathcal{I}\right).$$

Noticing the covariance is a constant matrix and KL-divergence between two Gaussians has closed-form solution; DDPM models $p_\theta(\mathbf{x}_{n-1} \mid \mathbf{x}_n) := \mathcal{N}(\mathbf{x}_{n-1} \mid \mu_\theta(\mathbf{x}_n, n), \frac{1-\bar{\alpha}_{n-1}}{1-\bar{\alpha}_n}\beta_n\mathcal{I})$. The mean vector $\mu_\theta(\mathbf{x}_n, n)$ is parameterized by a neural network.

The trick used in (Ho et al., 2020) is to reparameterize $\mu_\theta(\mathbf{x}_n, n)$ in terms of $\epsilon_\theta(\mathbf{x}_n, n)$ where it predicts the noise $\epsilon$ added to $\mathbf{x}_n$ from $\mathbf{x}_0$. (Note that $\mathbf{x}_n = \sqrt{\bar{\alpha}_n}\mathbf{x}_0 + \sqrt{1-\bar{\alpha}_n}\epsilon$ with $\epsilon \sim \mathcal{N}(0, \mathcal{I})$.)

Given this, the final loss function DDPM wants to minimize is:

$$\mathcal{L}_{\text{diff}} := \mathbb{E}_{n,\epsilon}\bigg[\|\epsilon_\theta\big(\sqrt{\bar{\alpha}_n}\mathbf{x}_0 + \sqrt{1-\bar{\alpha}_n}\epsilon, n\big) - \epsilon\|_2^2\bigg],$$

where the expectation is taken over $\epsilon \sim \mathcal{N}(0, \mathcal{I})$ and $n \sim \text{Unif}(\{0, \ldots, N\})$.

The generative model learns the *reverse process*. To generate new data from the learned distribution, the first step is to sample a point from the easy-to-sample distribution $\mathbf{x}_N \sim \mathcal{N}(0, \mathcal{I})$ and then iteratively denoise $(\mathbf{x}_N \to \mathbf{x}_{N-1} \to \cdots \to \mathbf{x}_0)$ it using the above model.

## G    EVALUATION METRIC

For the quantitative evaluation of synthesized data, we mainly focus on three criteria (1) the distributional similarities of the two tables; (2) the usefulness for predictive purposes; (3) the temporal and feature dependencies; We employ the following evaluation metrics:

***Discriminative Score*** (Yoon et al., 2019) measures the fidelity of synthetic time series data to original data, by training a classification model (optimizing a 2-layer LSTM) to distinguish between sequences from the original and generated datasets.

***Predictive Score*** (Yoon et al., 2019) measures the utility of generated sequences by training a posthoc sequence prediction model (optimizing a 2-layer LSTM) to predict next-step temporal vectors under a *Train-on-Synthetic-Test-on-Real* (TSTR) framework.

***Temporal Discriminative Score*** measures the similarity of distributions of *inter-row differences* between generated and original sequential data. This metric is designed to see if the generated data preserves the temporal dependencies of the original data. For any fixed integer $t \in \{1, \ldots, T-1\}$, the difference of the $n$-th row and $(n+t)$-th row in the table over $n \in \{1, \ldots, T-t\}$ is computed

for both generated and original data and discriminative score (Yoon et al., 2019) is computed over the differenced matrices from original and synthetic data. We average discriminative scores over 10 randomly selected $t \in \{1, \ldots, T-1\}$.

***Feature Correlation Score*** measures the averaged $L^2$-distance of correlation matrices computed on real and synthetic data. Following (Kotelnikov et al., 2022), to compute the correlation matrices, we use the Pearson correlation coefficient for numerical-numerical feature relationships, Theil's U statistics between categorical-categorical features, and the correlation ratio for categorical-numerical features. We use the following metrics to calculate the feature correlation score:

- **Pearson Correlation Coefficient**: Used for **Numerical** to **Numerical** feature relationship. Pearson's Correlation Coefficient $r$ is given by

$$r = \frac{\sum(x - \bar{x})(y - \bar{y})}{\sqrt{\sum(x - \bar{x})^2}\sqrt{\sum(y - \bar{y})^2}}$$

  where
  - $x$ and $y$ are samples in features $X$ and $Y$, respectively
  - $\bar{x}$ and $\bar{y}$ are the sample means in features $X$ and $Y$, respectively

- **Theil's U Coefficient**: Used for **Categorical** to **Categorical** feature relationship. Theil's U Coefficient $U$ is given by
$$U = \frac{H(X) - H(X|Y)}{H(X)}$$

  where
  - entropy of feature $X$ is defined as

$$H(X) = -\sum_x P_X(x) \log P_X(x)$$

  - entropy of feature $X$ conditioned on feature $Y$ is defined as

$$H(X|Y) = -\sum_{x,y} P_{X,Y}(x,y) \log \frac{P_{X,Y}(x,y)}{P_Y(y)}$$

  - $P_X$ and $P_Y$ are empirical PMF of $X$ and $Y$, respectively
  - $P_{X,Y}$ is the joint distribution of $X$ and $Y$

- **Correlation Ratio**: Used for **Categorical** to **Numerical** feature relationship. The correlation ratio $\eta$ is given by
$$\eta = \sqrt{\frac{\sum_x n_x (\bar{y}_x - \bar{y})^2}{\sum_{x,i} (y_{xi} - \bar{y})^2}}$$

  where
  - $n_x$ is the number of observations of label $x$ in the categorical feature
  - $y_{xi}$ is the $i$-th observation of the numerical feature with label $x$
  - $\bar{y}_x$ is the mean of observed samples $y_i \in Y$ with label $x$
  - $\bar{y}$ is the sample mean of $Y$

## H  SAMPLING OF THE LATENT MATRIX FROM (C)−TIMEAUTODIFF AND NETWORK ARCHITECTURE OF C−TIMEAUTODIFF

---

**Algorithm 1:** Sampling (Unconditional generation of TimeAutoDiff)

1 **Input: ts, t** $= \{t_1, \ldots, t_T\}$
2 $Z_N^{\text{Lat}} \sim \mathcal{N}(0, \mathcal{I}_{TF \times TF}).\text{reshape}(T, F)$
3 **while** $n = N, \ldots, 1$ **do**
4 $\quad$ **z** $\sim \mathcal{N}(0, \mathcal{I}_{TF \times TF}).\text{reshape}(T, F)$
5 $\quad Z_{n-1}^{\text{Lat}} = \frac{1}{\sqrt{\alpha_n}} \left( Z_n^{\text{Lat}} - \frac{1 - \alpha_n}{\sqrt{1 - \bar{\alpha}_n}} \epsilon_\theta(\mathbf{Z}_n^{\text{Lat}}, n, \mathbf{t}, \mathbf{ts}) \right) + \beta_n \mathbf{z}.\text{reshape}(T, F),$
6 **end**
7 **return** $Z_0^{\text{Lat}}.\text{reshape}(T, F)$

---

**Algorithm 2:** Sampling (Conditional generation of TimeAutoDiff)

1 **Input: ts, c, t** $= \{t_1, \ldots, t_T\}$
2 $Z_N^{\text{Lat}} \sim \mathcal{N}(0, \mathcal{I}_{TF \times TF}).\text{reshape}(T, F)$
3 **while** $n = N, \ldots, 1$ **do**
4 $\quad$ **z** $\sim \mathcal{N}(0, \mathcal{I}_{TF \times TF}).\text{reshape}(T, F)$
5 $\quad \tilde{\epsilon}_\theta := \epsilon_\theta(\mathbf{Z}_n^{\text{Lat}}, n, \mathbf{t}, \mathbf{ts}, \mathbf{c}),$
6 $\quad Z_{n-1}^{\text{Lat}} = \frac{1}{\sqrt{\alpha_n}} \left( Z_n^{\text{Lat}} - \frac{1 - \alpha_n}{\sqrt{1 - \bar{\alpha}_n}} \cdot \tilde{\epsilon}_\theta \right) + \beta_n \mathbf{z}.\text{reshape}(T, F),$
7 **end**
8 **return** $Z_0^{\text{Lat}}.\text{reshape}(T, F)$

---

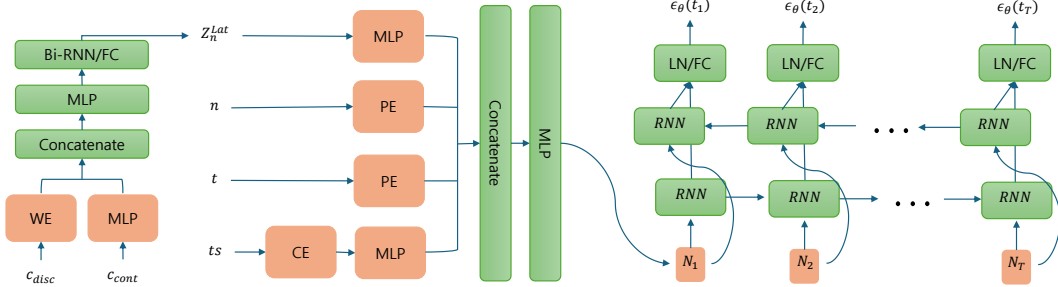

Figure 8: Network architecture of $\epsilon_\theta$ in C-TimeAutoDiff.

## I  GENERALIZABILITY OF TIMEAUTODIFF

In generative modeling, it is essential to check whether the learned model can generate the datasets not seen in the training set. If model memorizes and reproduces data points from the training dataset (Zhang et al., 2023b), this can undermine the primary motivation of data synthesizing: ***increasing dataset diversity***. To investigate further in this regard, we design an experiment using the notion of Distance to the Closest Record (DCR) (Park et al., 2018), which computes the Euclidean distance between a data point $r \in \mathbb{R}^{T \times F}$ in the synthesized dataset and the closest record to $r$ in the original table. We split the data into training (50%) / testing (50%) sets, where we only use the training set for model training.

**Interpretations of DCR scores:** DCR scores for both training and testing datasets can be used to evaluate the model's performance. Significant overlap between the DCR distributions of the training and testing datasets suggests that the model is drawing data from the data distribution, i.e., $\mathbb{P}(\mathbf{x}_1, \mathbf{x}_2, \ldots, \mathbf{x}_T)$. However, even with substantial overlap between the distributions, if the distances to the origin are small, this suggests that the patterns in the training and testing sets are alike, implying the model may have memorized specific training data points. If the DCR distribution of the training data is notably closer to zero compared to the testing data, it indicates that the model

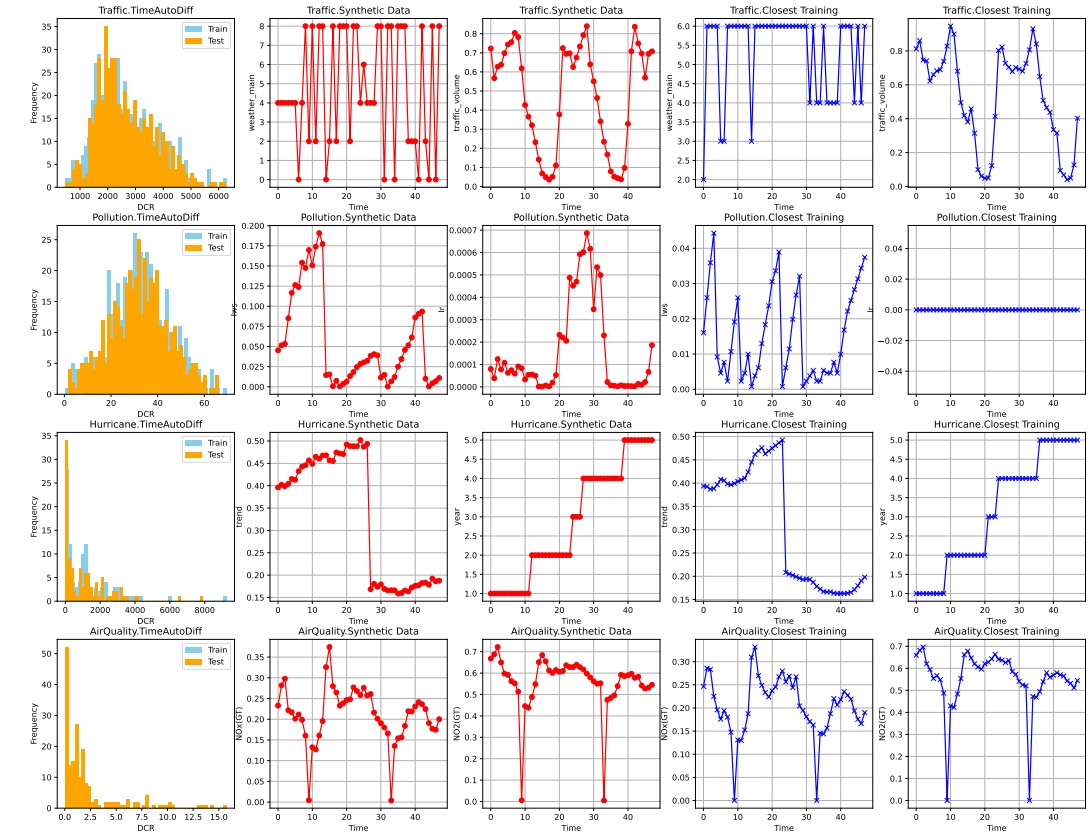

Figure 9: The leftmost column demonstrates that the DCR distributions for the training and testing sets exhibit significant overlap across four datasets (from top): *Traffic*, *Pollution*, *Hurricane*, and *AirQuality*. For each dataset, two variables are selected for visualizations. The second and third columns illustrate these chosen features over timestamps (sequence length = 48) for an arbitrary synthetic data point. The fourth and fifth columns present the same features for the closest data point in the training dataset. The model trained on the Traffic and Pollution datasets clearly generates new data points with distinct patterns, while the models trained on Hurricane and AirQuality datasets replicate their training data points, as indicated by DCR distributions being close to zero.

has memorized the training dataset. Last but not least, it's important to recognize that random noise can also produce similar DCR distributions. Therefore, the DCR score should be evaluated in conjunction with other measures of fidelity, such as the discriminative score, and utility measures, such as the predictive score, to provide a comprehensive assessment of the model's generalization capabilities. We provide the interpretations of DCR distributions of `TimeAutoDiff` for *Traffic*, *Pollution*, *Hurricane*, and *AirQuality* datasets in the caption of Fig. 9.

## J VOLATILITY AND MOVING AVERAGE: COMPARISON BETWEEN REAL AND SYNTHETIC UNDER STOCK DATA

We provide the performance of our model in terms of volatility and moving average. We first provide the brief descriptions on Simple Moving Average, Exponential Moving Average, and Volatility.

**Simple Moving Average (SMA)**: The Simple Moving Average (SMA) is computed as the arithmetic mean of values over a sliding window of size $w = 5$. For a given time step $t$, the SMA is given by:

$$\text{SMA}_t = \frac{1}{5} \sum_{i=t-4}^{t} \text{Value}_i$$

where $\text{Value}_i$ represents the value of the time series at time $i$. This metric smooths short-term fluctuations and highlights the overall trend by averaging values in the specified window.

**Exponential Moving Average (EMA)** The Exponential Moving Average (EMA) is a weighted average of values where recent data points have exponentially greater weight. For a window size of $w = 5$, the smoothing factor $\alpha$ is computed as: $\alpha = \frac{2}{w+1} = \frac{2}{5+1} = \frac{1}{3}$. The EMA at time $t$ is then computed recursively as:

$$\text{EMA}_t = \alpha \cdot \text{Value}_t + (1 - \alpha) \cdot \text{EMA}_{t-1}$$

where $\text{Value}_t$ is the current value of the time series, and $\text{EMA}_{t-1}$ is the EMA from the previous time step. This method emphasizes recent changes while retaining some information from the historical trend.

**Volatility** Volatility measures the degree of variation in the time series over a sliding window of size $w = 5$. It is calculated as the rolling standard deviation of the percentage changes (returns). First, the percentage change (return) between consecutive values is computed as:

$$\text{Return}_i = \frac{\text{Value}_i - \text{Value}_{i-1}}{\text{Value}_{i-1}}$$

For a given time step $t$, the volatility over the window $w = 5$ is given by: $\text{Volatility}_t = \sqrt{\frac{1}{5} \sum_{i=t-4}^{t} \left( \text{Return}_i - \bar{\text{Return}} \right)^2}$ where $\bar{\text{Return}}$ is the mean of the returns within the window.

**Results:** We work on the stock data. The figure 12 provide a clear side-by-side comparison between the synthetic and real data, with the left column displaying the synthetic data and the right column showcasing the corresponding real data for two selected features (Open & Close prices) over 200 timestamps (i.e.,T=200). Each row focuses on one feature, allowing for a detailed examination of the behavior across key metrics: Simple Moving Average (SMA), Exponential Moving Average (EMA), and Volatility. The SMA and EMA curves, plotted alongside the raw time series data, highlight the ability of the synthetic data to replicate the long-term trends (SMA) and short-term responsiveness (EMA) observed in the real data. Volatility, overlaid as a secondary y-axis in each plot, demonstrates the synthetic data's capacity to reproduce the temporal variability, including periods of high and low uncertainty, as reflected in the real data. The remarkable alignment across all metrics suggests that the synthetic data closely mirrors the real data's dynamics, effectively capturing both the overall patterns and nuanced fluctuations. This visual comparison underscores the robustness and reliability of the synthetic data generation process.

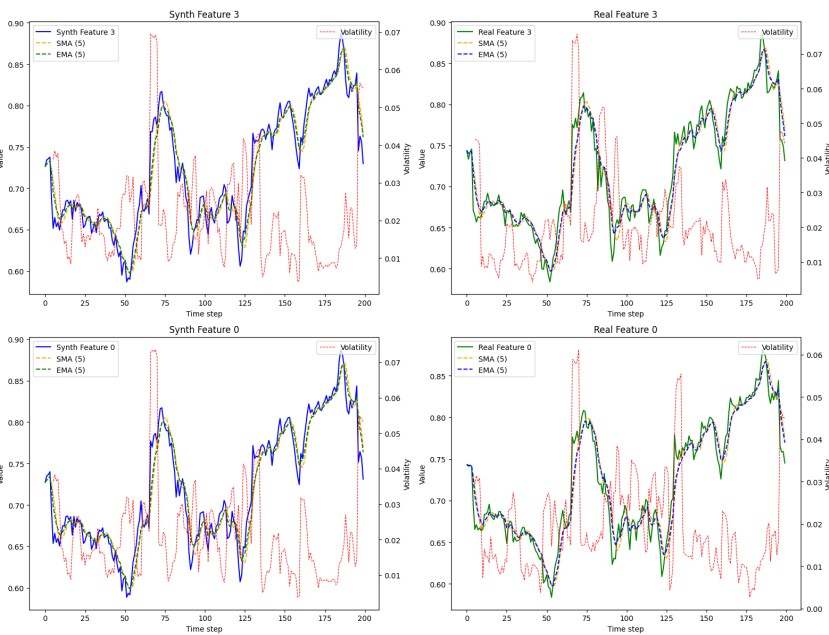

Figure 10: Comparison of synthetic (left) and real (right) data across two features, illustrating alignment in trends (SMA, EMA) and variability (Volatility: secondary y-axis).

## K MODEL PARAMETER SETTINGS, TRAINING & HYPER-PARAMETER CHOICES

Our model consists of two components: **VAE** and **DDPM**. We present the sizes of networks in both components that are applied entirely across the experiments in the paper.

$$\text{VAE-Encoder} = \big\{ \text{Dimension of first FC-layer in MLP-block for encoded features:}$$
$$(\text{Num of disc var.} \times \mathbf{128} + \text{Num of cont var.} \times \mathbf{16}) \times \mathbf{128},$$
$$\text{Dimension of second FC-layer in MLP-block for encoded features:} \mathbf{128} \times \mathbf{F},$$
$$\text{Dimension of hidden layer for the 2-RNNs for } \mu \text{ and } \sigma: \mathbf{200},$$
$$\text{Number of layers for the 2-RNNs for } \mu \text{ and } \sigma: \mathbf{2},$$
$$\text{Dimension of fully-connected layer topped on 2-RNNs: } \mathbf{200} \times \mathbf{F} \big\}$$

$$\text{VAE-Decoder} = \big\{ \text{Dimension of first FC-layer in MLP-block for latent matrix } \mathbf{Z}_0: \mathbf{F} \times \mathbf{128},$$
$$\text{Dimension of second FC-layer in MLP-block for latent matrix } \mathbf{Z}_0: \mathbf{128} \times \mathbf{128} \big\}$$

$$\text{DDPM} = \big\{ \text{Output dimensions of encodings of } (\mathbf{Z}_n^{\text{Lat}}, n, \mathbf{t}, \mathbf{ts}): \mathbf{200},$$
$$\text{Dimension of hidden layer for the Bi-RNNs: } \mathbf{200},$$
$$\text{Number of layers for the Bi-RNNs: } \mathbf{2},$$
$$\text{Dimension of FC-layer of the output of Bi-RNNs: } \mathbf{400} \times \mathbf{F},$$
$$\text{Diffusion Steps: } 100 \big\}$$

Training for both the VAE and DDPM models is set to 25,000 epochs. The batch size for VAE training is 100, while the batch size for DDPM training matches the number of diffusion steps. We use the Adam optimizer, with a learning rate of $2 \times 10^{-4}$ decaying to $10^{-6}$ for the VAE, and a learning rate of $10^{-3}$ for the DDPM. For stabilization of diffusion training, we employ Exponential Moving Average (EMA) with decay rate 0.995. We employ linear noise scheduling for $\beta_n := 1 - \alpha_n$, $n \in \{1, 2, \ldots, N\}$ with $\beta_1 = 10^{-4}$ and $\beta_N = 0.2$:

$$\beta_n = \left( 1 - \frac{n}{N} \right) \beta_1 + \frac{n}{N} \beta_N.$$

In the following, we investigate the robustness of our models to the various hyper-parameter choices in VAE and DDPM. Specifically, we studied the effects of (1) feature dimension of $Z_0^{\text{Lat}}$ ($F/2, F/4$), (2) number of diffusion steps $(75, 50, 25)$, (3) training epochs of VAE and DDPM $(20000, 15000, 10000)$, (4) dimension of hidden layers of two RNNs (for $\mu$ and $\sigma$) in VAE $(150, 100, 50)$, (5) dimension of hidden layers of Bi-RNNs in DDPM $(150, 100, 50)$, (6) the number of layers of two RNNs (for $\mu$ and $\sigma$) in VAE $(1)$, (7) the number of layers of Bi-RNNs in DDPM $(1)$. (8) the quadratic noise scheduler used in Song et al. (2020a); Tashiro et al. (2021):

$$\beta_n = \left( \left( 1 - \frac{n}{N} \right) \sqrt{\beta_1} + \frac{n}{N} \sqrt{\beta_N} \right)^2.$$

with the minimum noise level $\beta_1 = 0.0001$, and the maximum noise level $\beta_N = 0.5$.

The experiments are conducted over the varying parameters (in the paranthesis), while the remaining parameters in the model are being fixed as in the above settings. The first 2000 rows of **_Traffic_** data are used for the experiments with sequence length 24. (i.e., the dimensions of tensors used in the experiments are [B, T, F ] = [1977, 24, 8])

**Results Interpretations:** Table 5 presents the performance of the models across four metrics, with variations in hyperparameter settings. Overall, larger models yield better results. Reducing the diffusion steps, dimensions, and the number of hidden layers in RNNs within the VAE and Bi-RNN components of DDPM significantly degrades model performance. Longer training of both VAE and DDPM consistently enhances results. The linear noise scheduler outperforms the quadratic noise scheduler. While reducing the feature dimension to $F/2$ slightly improves discriminative and temporal discriminative scores, further compression to $F/4$ leads to information loss during signal reconstruction, resulting in poorer performance.

| Method | Disc. Score | Pred. Score | Temp. Disc Score | Feat. Correl. |
|---|---|---|---|---|
| TimeAutoDiff | 0.015(0.012) | 0.229(0.010) | 0.034(0.020) | 0.043(0.000) |
| Latent Feature Dimension = F/2 | 0.009(0.004) | 0.227(0.009) | 0.096(0.061) | 0.055(0.000) |
| Latent Feature Dimension = F/4 | 0.038(0.021) | 0.233(0.007) | 0.099(0.171) | 0.048(0.000) |
| Diffusion Steps = 75 | 0.016(0.009) | 0.224(0.015) | 0.014(0.009) | 0.039(0.000) |
| Diffusion Steps = 50 | 0.118(0.019) | 0.241(0.003) | 0.092(0.046) | 0.109(0.000) |
| Diffusion Steps = 25 | 0.150(0.027) | 0.248(0.006) | 0.111(0.065) | 0.100(0.000) |
| VAE Training = 15000 | 0.075(0.009) | 0.243(0.005) | 0.035(0.007) | 0.091(0.000) |
| VAE Training = 10000 | 0.068(0.018) | 0.242(0.007) | 0.038(0.038) | 0.050(0.000) |
| VAE Training = 5000 | 0.195(0.025) | 0.245(0.002) | 0.039(0.019) | 0.077(0.000) |
| DDPM Training = 15000 | 0.098(0.014) | 0.237(0.015) | 0.062(0.038) | 0.086(0.000) |
| DDPM Training = 10000 | 0.220(0.025) | 0.246(0.004) | 0.165(0.045) | 0.195(0.000) |
| DDPM Training = 5000 | 0.267(0.021) | 0.255(0.001) | 0.216(0.031) | 0.190(0.000) |
| Hidden Dimension of RNNs (VAE) = 150 | 0.013(0.008) | 0.240(0.007) | 0.031(0.009) | 0.015(0.000) |
| Hidden Dimension of RNNs (VAE) = 100 | 0.030(0.009) | 0.236(0.017) | 0.017(0.011) | 0.039(0.000) |
| Hidden Dimension of RNNs (VAE) = 50 | 0.082(0.023) | 0.238(0.004) | 0.051(0.038) | 0.064(0.000) |
| Hidden Dimension of Bi-RNNs (DDPM) = 150 | 0.031(0.010) | 0.243(0.011) | 0.028(0.013) | 0.035(0.000) |
| Hidden Dimension of Bi-RNNs (DDPM) = 100 | 0.167(0.012) | 0.248(0.003) | 0.094(0.054) | 0.119(0.000) |
| Hidden Dimension of Bi-RNNs (DDPM) = 50 | 0.174(0.014) | 0.251(0.005) | 0.157(0.072) | 0.132(0.000) |
| Number of layers in RNNs (VAE) = 1 | 0.024(0.013) | 0.245(0.009) | 0.042(0.018) | 0.028(0.000) |
| Number of layers in Bi-RNNs (DDPM) = 1 | 0.097(0.009) | 0.250(0.002) | 0.245(0.009) | 0.086(0.000) |
| Quadratic Noise Scheduler | 0.109(0.017) | 0.234(0.013) | 0.072(0.025) | 0.106(0.000) |

Table 5: Performances measured with various choices of hyper-parameters in `TimeAutoDiff`. The experiments are conducted on Traffic dataset with $T = 24$.

## L    RESULTS ON ABLATION TEST, $\beta$-SCHEDULING & SCALABILITY

**Ablation:**  The ablation test results are summarized in Table 7. A single model alone (i.e., only VAE or DDPM) cannot accurately capture the statistical properties of the distributions of tables, which strongly supports the motivation of our model. The components related to the diffusion model, such as timestamp encoding and Bi-RNN, impact the generative performance across most cases as models lacking these components do not exhibit optimal performance. The encodings for continuous features in the VAE notably enhance the fidelity and temporal dependences of the generated data.

Additionally, we consider the following scenarios:

1. Replacing the MLP with an RNN in the decoder of the VAE.

2. Replacing the two RNNs with an MLP in the encoder of the VAE.

3. Inspired by Biloš et al. (2023), we explore injecting continuous noise from a stochastic process (Gaussian process) into the DDPM. Specifically, the perturbation kernel

$$q(\mathbf{z}_{n,j}^{\text{Lat}}|\mathbf{z}_{0,j}^{\text{Lat}}) = \mathcal{N}(\sqrt{\bar{\alpha}_n}\mathbf{z}_{0,j}^{\text{Lat}}, (1 - \bar{\alpha}_n)\mathbf{\Sigma})$$

   is applied independently to each column of $\mathbf{Z}_0^{\text{Lat}} \in \mathbb{R}^{T \times F}$, where $\mathbf{\Sigma}_{ij} = \exp(-\gamma|\mathbf{t}_i - \mathbf{t}_j|)$ with $\gamma = 0.2$.

The experimental results show that none of the ablated models outperformed the original configuration significantly. Specifically, the second configuration demonstrates the benefits of modeling temporal relations twice in the VAE and DDPM due to the following reasons:

**Hierarchical Temporal Dependency Modeling:** The VAE encoder captures compact latent representations with temporal dependencies, providing a structured input for the diffusion model. This allows the diffusion process to refine finer-grained patterns without redundantly encoding high-level temporal structures, resulting in more realistic outputs.

**Noise-Tolerant Latent Representation:** Encoding temporal dependencies early in the VAE encoder ensures that the latent variable $\mathbf{z}$ is robust to noise. This noise resilience helps maintain critical temporal structures during the diffusion process, enhancing the fidelity of the generated data.

**The effect of adaptive $\beta$-VAE:** Motivated from (Zhang et al., 2023a), we evaluate the effects of scheduling on $\beta$ coefficients in VAE in terms of tradeoffs between reconstruction error and KL-divergence. In Fig 11, we observe that while large $\beta$ can ensure the close distance between the embedding and standard normal distributions, its reconstruction loss is relatively larger than that of smaller ones, and vice versa. The adaptive $\beta$-scheduling ensures both the lowest reconstruction error

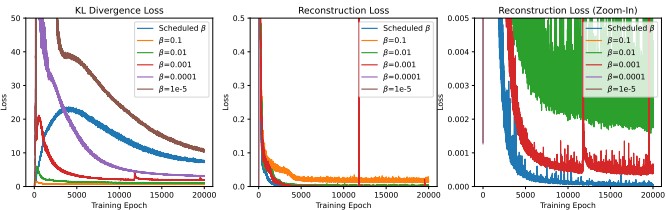

| $\beta$ | Disc. Score |
|---|---|
| $10^{-1}$ | $0.369(0.101)$ |
| $10^{-2}$ | $0.041(0.011)$ |
| $10^{-3}$ | $0.043(0.019)$ |
| $10^{-4}$ | $0.079(0.012)$ |
| $10^{-5}$ | $0.043(0.009)$ |
| Scheduled $\beta$ | $\mathbf{0.023(0.015)}$ |

Figure 11: KL-Divergence (left) and Reconstruction (middle) losses over 20000 training iterations of VAE on Traffic dataset. The zoomed-in panel (right) displays the scheduled-$\beta$ reaches the lowest reconstruction error stably without any spikes.

Table 6: The results of discriminative scores with varying $\beta$ values on the Traffic dataset.

and relatively lower KL-divergence, preserving the shape of embedding distribution. The adaptive $\beta$-scheduling achieves the fastest and the most stable signal reconstructions among other $\beta$-choices. Table 6 shows the effectiveness of $\beta$-scheduling for quality of synthetic data in discriminative score.

**Scalability:** We investigate the scalability of `TimeAutoDiff` by varying the sequence length (i.e., $T$) and the number of features (i.e., $F$). For the experiment, we follow the sine wave synthetic setting in TimeGAN paper (Yoon et al., 2019).

***Sine Waves.*** We simulate multivariate sinusoidal sequences of different frequencies $\eta$ and phases $\theta$, providing continuous-valued, periodic, multivariate data where each feature is independent of others. For each dimension $i \in \{1, ..., F\}$, $x_i(t) = \sin(2\pi\eta t + \theta)$, where $\eta \sim \text{Unif}[0, 1]$ and $\theta \sim \text{Unif}[-\pi, \pi]$.

We train the model with data of size [Batch Size $\times$ Seq Len $\times$ Feature Dim] and draw the samples with same sizes. In the following Tables, training time for VAE, Diffusion models, and sampling time for data are recorded in seconds. Allocated GPU memory for sampling (in MB), discriminative score and temporal discriminative score are also recorded.

Under the model configurations stated in the Appendix K, `TimeAutoDiff` can generate the sequence of length 900 with 5 features with good fidelities. (See Table 8.) In contrast, we observe a performance drop when the feature sizes increase (30 to 50 features) with a sequence length of 200. To address this, we reduce the dimension of the feature axis in the latent space to $F/2$, resulting in a significant performance increase in the high-dimensional feature setting.

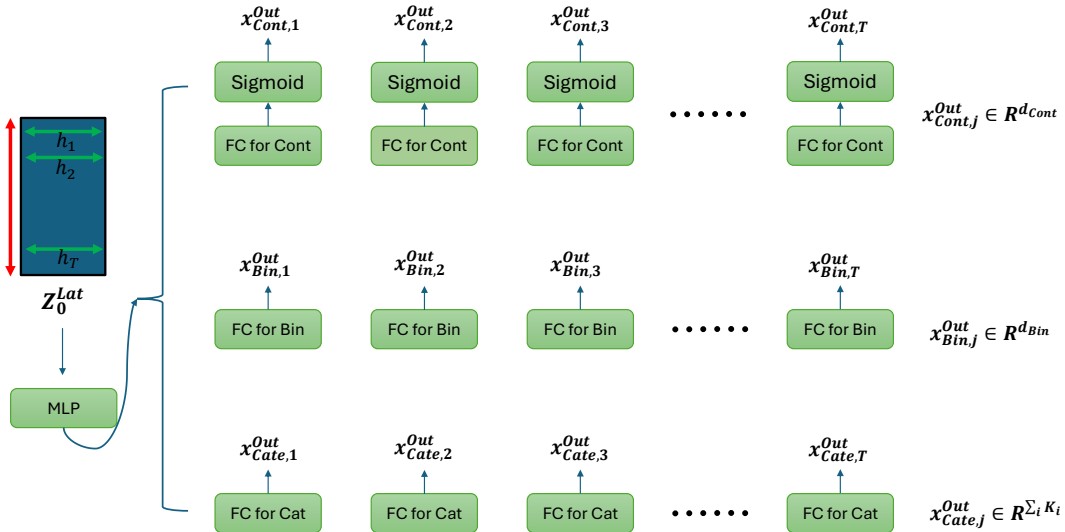

Figure 12: Decoder has a simple design: the latent matrix $\mathbf{Z}_0^{\text{Lat}}$ is fed to a shared MLP blcock, and the output of the MLP block is fed to the different linear layers based on the data type. Sigmoid function is used to match the scale of continuous input data.

| Metric | Method | Traffic | Pollution | Hurricane | AirQuality |
|---|---|---|---|---|---|
| Discriminative Score (The lower, the better) | TimeAutoDiff | 0.027(0.014) | **0.014(0.011)** | **0.035(0.010)** | 0.035(0.016) |
| | only VAE | 0.476(0.010) | 0.491(0.010) | 0.490(0.010) | 0.494(0.007) |
| | only DDPM | 0.283(0.131) | 0.313(0.163) | 0.252(0.034) | 0.266(0.048) |
| | w/o Encoding equation 1 | 0.029(0.017) | 0.062(0.015) | 0.063(0.018) | 0.072(0.020) |
| | w/o Timestamps | 0.095(0.016) | 0.105(0.012) | 0.171(0.085) | 0.074(0.013) |
| | w/o Bi-directional RNN | 0.049(0.015) | 0.021(0.020) | 0.300(0.036) | **0.019(0.015)** |
| | RNN in decoder (VAE) | 0.186(0.019) | 0.185(0.020) | 0.198(0.031) | 0.124(0.018) |
| | MLP in encoder (VAE) | 0.017(0.011) | 0.072(0.020) | 0.117(0.019) | 0.067(0.025) |
| | Smooth Noise (DDPM) | **0.015(0.009)** | 0.078(0.013) | 0.140(0.016) | 0.140(0.016) |
| Predictive Score (The lower, the better) | TimeAutoDiff | 0.229(0.010) | **0.008(0.000)** | 3.490(0.097) | **0.004(0.000)** |
| | only VAE | 0.241(0.001) | 0.008(0.000) | 4.566(0.041) | 0.019(0.002) |
| | only DDPM | 0.241(0.012) | 0.016(0.000) | **0.034(0.007)** | 0.009(0.002) |
| | w/o Encoding equation 1 | **0.219(0.011)** | 0.008(0.000) | 3.611(0.216) | 0.005(0.000) |
| | w/o Timestamps | 0.241(0.003) | 0.008(0.000) | 4.228(0.248) | 0.004(0.000) |
| | w/o Bi-directional RNN | 0.231(0.008) | 0.008(0.000) | 3.549(0.047) | 0.004(0.000) |
| | RNN in decoder (VAE) | 0.232(0.008) | 0.008(0.000) | 3.598(0.095) | 0.012(0.004) |
| | MLP in encoder (VAE) | 0.220(0.011) | 0.008(0.000) | 3.365(0.072) | 0.061(0.002) |
| | Smooth Noise (DDPM) | 0.221(0.011) | 0.008(0.000) | 0.091(0.027) | 0.059(0.001) |
| Temporal Discriminative Score (The lower, the better) | TimeAutoDiff | 0.047(0.017) | **0.008(0.005)** | **0.020(0.010)** | 0.035(0.024) |
| | only VAE | 0.368(0.107) | 0.484(0.043) | 0.490(0.014) | 0.493(0.006) |
| | only DDPM | 0.197(0.127) | 0.135(0.131) | 0.213(0.096) | 0.242(0.122) |
| | w/o Encoding equation 1 | 0.036(0.016) | 0.052(0.019) | 0.049(0.022) | **0.008(0.005)** |
| | w/o Timestamps | 0.084(0.047) | 0.053(0.018) | 0.117(0.065) | 0.064(0.019) |
| | w/o Bi-directional RNN | 0.031(0.021) | 0.047(0.057) | 0.404(0.013) | 0.023(0.015) |
| | RNN in decoder (VAE) | 0.130(0.025) | 0.133(0.019) | 0.324(0.072) | 0.331(0.130) |
| | MLP in encoder (VAE) | 0.037(0.017) | 0.060(0.018) | 0.094(0.019) | 0.045(0.032) |
| | Smooth Noise (DDPM) | **0.020(0.007)** | 0.059(0.029) | 0.090(0.027) | 0.091(0.027) |
| Feature Correlation Score (The lower, the better) | TimeAutoDiff | **0.022(0.014)** | **1.104(0.900)** | **0.069(0.027)** | **0.147(0.230)** |
| | only VAE | 0.404(0.339) | 1.329(0.757) | 0.427(0.371) | 0.702(1.001) |
| | only DDPM | 2.238(1.530) | 2.020(1.460) | 2.380(1.513) | 0.198(0.298) |
| | w/o Encoding equation 1 | 0.029(0.021) | 1.148(0.850) | 0.077(0.034) | 0.266(0.405) |
| | w/o Timestamps | 0.247(0.521) | 1.303(0.793) | 0.097(0.044) | 0.231(0.349) |
| | w/o Bi-directional RNN | 0.048(0.024) | 1.227(0.863) | 0.090(0.043) | 0.155(0.256) |
| | RNN in decoder (VAE) | 0.413(0.544) | 1.187(0.820) | 0.247(0.123) | 0.913(1.302) |
| | MLP in encoder (VAE) | 0.025(0.015) | 1.240(0.853) | 0.122(0.058) | 1.217(1.745) |
| | Smooth Noise (DDPM) | 0.059(0.037) | 1.246(0.843) | 0.882(1.271) | 1.215(1.345) |

Table 7: The experimental results of ablation test in `TimeAutoDiff`. The bolded number indicates the best-performing model.

| Batch Size | Seq Len | VAE | Diff | Sampling | GPU Mem | Disc Scr | Temp Disc Scr |
|---|---|---|---|---|---|---|---|
| 500 | 100 | 187.23 | 94.32 | 1.294 | 910.47 | 0.067 (0.034) | 0.143 (0.114) |
| 400 | 300 | 420.23 | 201.47 | 3.585 | 1991.75 | 0.040 (0.023) | 0.064 (0.059) |
| 300 | 500 | 665.69 | 315.92 | 5.511 | 2572.63 | 0.032 (0.016) | 0.078 (0.078) |
| 200 | 700 | 928.83 | 415.36 | 7.303 | 2466.91 | 0.048 (0.016) | 0.193 (0.122) |
| 100 | 900 | 1209.34 | 530.36 | 8.499 | 1670.75 | 0.16 (0.094) | 0.13 (0.143) |

Table 8: The number of feature is fixed as $5$. The sequence length increases up to $900$.

| Batch Size | Feat Dim | VAE | Diff | Sampling | GPU Mem | Disc Scr | Temp Disc Scr |
|---|---|---|---|---|---|---|---|
| 800 | 10 | 128.11 | 355.03 | 5.36 | 4696.21 | 0.24 (0.08) | 0.26 (0.09) |
| 800 | 20 | 132.48 | 359.32 | 4.12 | 5080.84 | 0.26 (0.05) | 0.38 (0.08) |
| 800 | 30 | 134.02 | 371.38 | 3.99 | 5540.96 | 0.31 (0.08) | 0.33 (0.17) |
| 800 | 40 | 134.72 | 364.85 | 3.97 | 6003.00 | 0.39 (0.14) | 0.41 (0.14) |
| 800 | 50 | 135.95 | 374.61 | 5.35 | 6464.41 | 0.48 (0.02) | 0.49 (0.00) |

Table 9: The sequence length is fixed as $200$. The feature dimension increases up to $50$.

| Batch Size | Feat Dim | VAE | Diff | Sampling | GPU Mem | Disc Scr | Temp Disc Scr |
|---|---|---|---|---|---|---|---|
| 800 | 10 | 131.65 | 365.81 | 4.63 | 3288.57 | 0.20 (0.12) | 0.23 (0.14) |
| 800 | 20 | 128.34 | 344.86 | 4.53 | 3947.75 | 0.25 (0.13) | 0.29 (0.09) |
| 800 | 30 | 130.92 | 363.41 | 4.61 | 4358.76 | 0.17 (0.11) | 0.34 (0.14) |
| 800 | 40 | 132.03 | 359.15 | 4.58 | 4771.51 | 0.24 (0.19) | 0.38 (0.09) |
| 800 | 50 | 134.96 | 367.05 | 4.70 | 5185.07 | 0.32 (0.18) | 0.41 (0.10) |

Table 10: Same setting with Table 9, but the dimension of latent matrix is set as $200 \times 7$.

# M    ADDITIONAL PLOTS: AUTO-CORRELATION / PERIODIC, CYCLIC PATTERNS

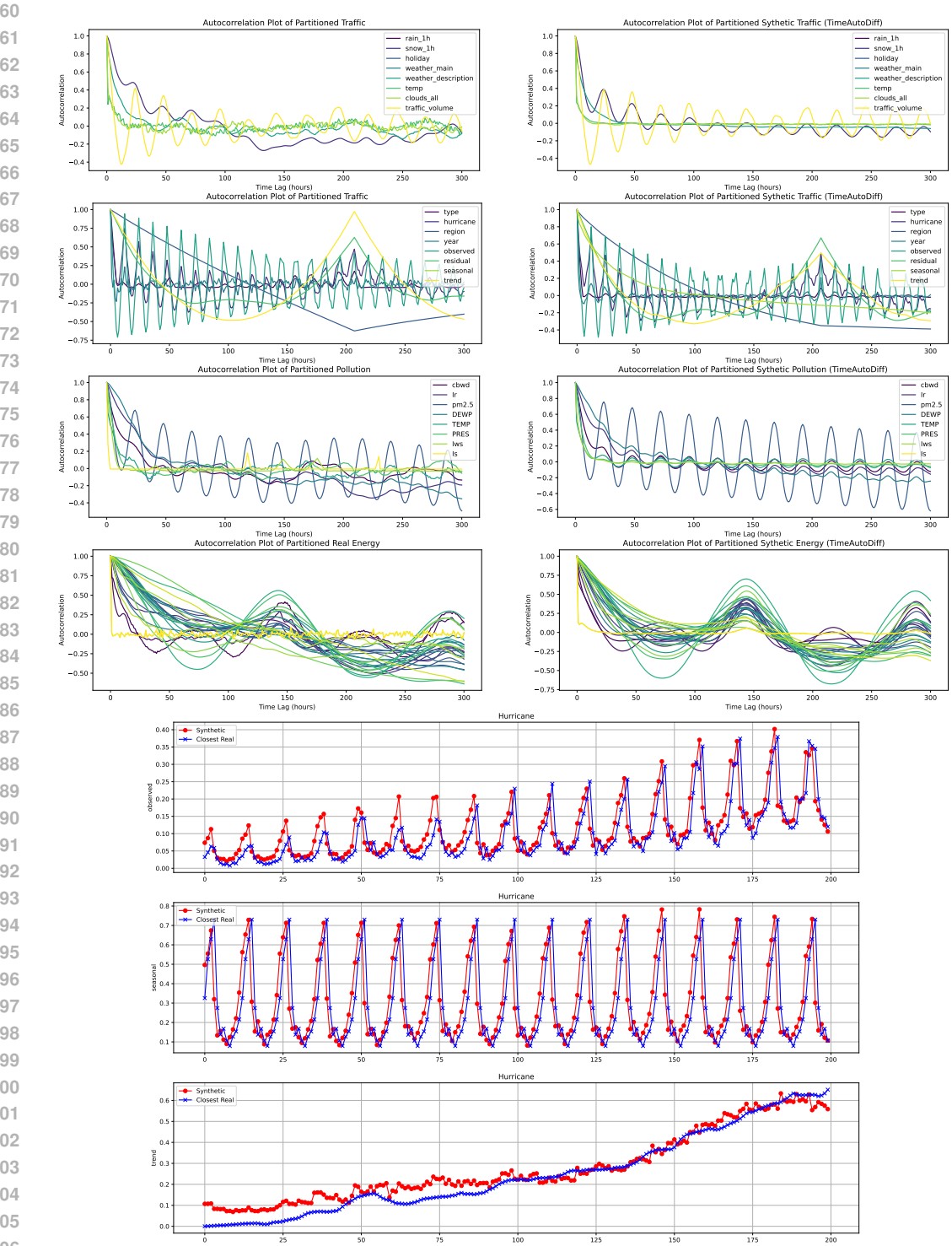

Figure 13: The first four plots from the top are auto-correlation plots of lag 300 for real (left) and synthetic (right) of 'Traffic','Hurricane','Pollution', and 'Energy'. The last three plots are ['Observed', 'Seasonal', 'Trend'] variables of Hurricane dataset The sequence length of generated synthetic data for AC (first four) and cyclic / trend pattern (last three) are 500 and 200, respectively.

# N   METADATA CONDITIONAL GENERATION FROM C-TIMEAUTODIFF

`C-TimeAutoDiff` can conditionally generate heterogeneous outputs that include both categorical and continuous variables.

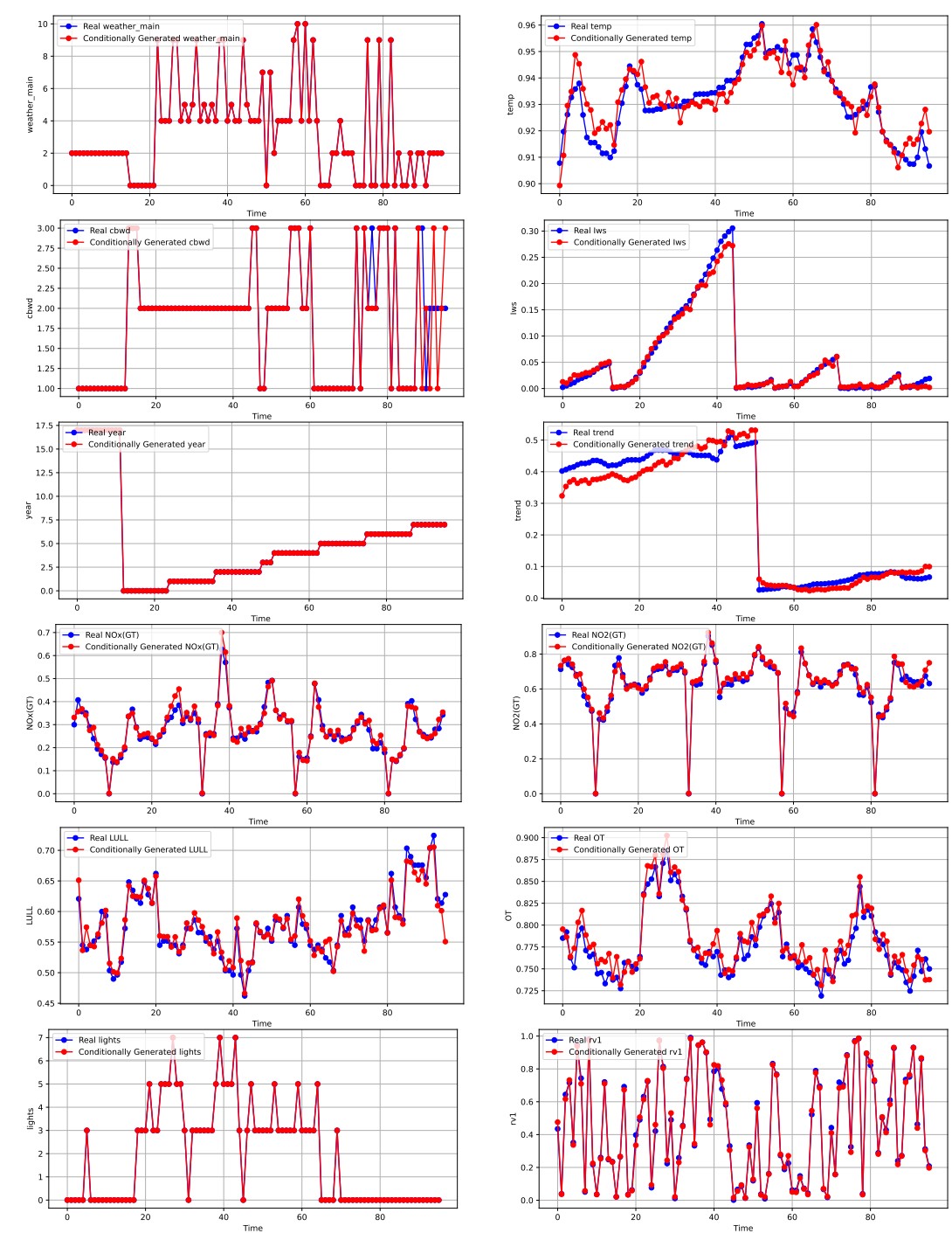

Figure 14: Datasets: (output variables) from top to bottom: ***Traffic***: ('Weather main', 'temp'), ***Pollution***: ('cbwd', 'Iws'), ***Hurricane***: ('year', 'trend'), ***AirQuality***: ('NOx(GT)', 'NO2(GT)'), ***ETTh1***: ('LULL', 'OT'), ***Energy***: ('lights', 'rv1'). The output is chosen to be heterogeneous (except AirQuality & ETTh1) both having discrete and continuous variables. Conditional variables **c** are set as remaining variables from the entire features. See the list of entire features of each dataset through the link in Appendix C.

# O    MAXIMUM MEAN DISCREPANCY & ENTROPY

We used two metrics proposed by TSGM (Nikitin et al., 2023): Maximum Mean Discrepancy (MMD) and Entropy. MMD measures the similarity (or fidelity) between synthetic and real time series data, while Entropy assesses the diversity of the synthetic data. The results are summarized in Table 11 and are consistent with those in Table 1.

TimeAutoDiff achieves the lowest MMD scores across all four datasets, aligning with the discriminative scores reported in Table 1. This indicates that TimeAutoDiff effectively generates synthetic data that closely resembles real data. For diversity, higher Entropy values indicate a dataset with more diverse samples. However, as noted in (Nikitin et al., 2023), Entropy should be considered alongside other metrics, as random noise can also result in high Entropy values. TimeAutoDiff produces synthetic data with higher Entropy values than the real data, though not as excessively as other baseline models. This suggests that our model generates synthetic data that preserves the statistical properties of the original data, maintaining diversity without introducing excessive deviation.

| Metric | Method | Traffic | Pollution | Hurricane | AirQuality |
|---|---|---|---|---|---|
| MMD Score

(The lower, the better) | TimeAutoDiff | 0.000629 | 0.000895 | 0.000891 | 0.001531 |
| | TimeGAN | 0.001738 | 0.009791 | 0.002775 | 0.042986 |
| | DoppelGANer | 0.000644 | 0.000960 | 0.005489 | 0.017038 |
| | Diffusion-TS | 0.005099 | 0.037102 | 0.078387 | 0.004144 |
| | TSGM | 0.001484 | 0.006322 | 0.031971 | 0.013777 |
| | real vs. real | 0.000000 | 0.000000 | 0.000000 | 0.000000 |
| Entropy Score

(Needs to be considered with other metrics) | TimeAutoDiff | 6419.404 | 8472.642 | 7129.152 | 16570.016 |
| | TimeGAN | 6714.156 | 11021.597 | 7804.343 | 15343.967 |
| | DoppelGANer | 3941.083 | 8656.403 | 6946.678 | 8708.616 |
| | Diffusion-TS | 9763.042 | 7372.591 | 9861.151 | 15934.365 |
| | TSGM | 11899.225 | 11854.764 | 6535.306 | 15766.673 |
| | Real | 5983.576 | 6976.253 | 6613.284 | 14952.996 |

Table 11: Maximum Mean Discrepancy (MMD) and Entropy of TimeAutoDiff, TimeGAN, DoppelGANer, Diffusion-TS, TSGM and Real data. The experimental setting is same with that of Table ??.

