# OpenReview forum: "TimeAutoDiff: Generation of Heterogeneous Time Series Data via Latent Diffusion Model"
_ICLR.cc/2025/Conference — ICLR 2025 Conference Withdrawn Submission_

### Official Review · Reviewer_Q7zx · 2024-11-02

**Soundness:** 3
**Presentation:** 4
**Contribution:** 3
**Rating:** 3
**Confidence:** 4

**Summary:**

The paper introduces TimeAutoDiff, a time series tabular data synthesizer that combines the Variational Auto-Encoder (VAE) and Denoising Diffusion Probabilistic Model (DDPM). It effectively manages heterogeneous features and enhances data generation fidelity and temporal dependencies. The model integrates a latent diffusion framework with a specialized VAE, improving performance in high-dimensional settings. It supports applications such as missing data imputation, privacy, and interpretability, and demonstrates superior results compared to existing models in generating synthetic time series data.

**Strengths:**

The paper is easy to read and the experiments are comprehensive.

**Weaknesses:**

One concern is its current inability to generate interpretable results. In many practical applications, especially in high-stakes fields like finance and healthcare, stakeholders must clearly understand how models make decisions. The lack of interpretability can lead to skepticism and reluctance to adopt the model, as users may hesitate to trust a system whose inner workings are opaque. This limitation could hinder the model's applicability in scenarios where understanding the rationale behind generated data is crucial for decision-making; Another point is the pure focus on continuous data: While the method demonstrates good performance with continuous data, its methods are primarily tailored for this data type. This focus raises concerns about the model's effectiveness when dealing with heterogeneous datasets that include categorical features. The inability to seamlessly integrate and process diverse data types could restrict the model's usability in real-world applications where such heterogeneity is common; another point is the dependence on latent space reduction: The paper notes a performance drop when the feature sizes increase, which necessitated a reduction in the latent space dimension. This reliance on dimensionality reduction to maintain performance raises concerns about the model's scalability and robustness. If the model's effectiveness is sensitive to the dimensionality of the latent space, it may struggle to perform well in high-dimensional settings without careful tuning.

**Questions:**

1, How can the model be made more robust to high-dimensional feature spaces without relying heavily on latent space reduction?

2, What modifications or extensions to the framework would be necessary to effectively handle heterogeneous datasets that include both continuous and categorical features?

---

> ### Author Response · Authors · 2024-11-24
> **Response to the comments from Reviewer Q7zx (1)**
>
> Thanks for your comments on our work! We made significant revisions inspired by your feedback. Please take a look at our revised manuscript and followings are the answers to your questions and comments on weakness and questions:
>
> Q) One concern is its current inability to generate interpretable results. In many practical applications, especially in high-stakes fields like finance and healthcare, stakeholders must clearly understand how models make decisions. The lack of interpretability can lead to skepticism and reluctance to adopt the model, as users may hesitate to trust a system whose inner workings are opaque. This limitation could hinder the model's applicability in scenarios where understanding the rationale behind generated data is crucial for decision-making;
>
> A) This comment is exactly consistent with what we have mentioned in Appendix A, discussing the future directions of our work. Specifically, as mentioned in the “Interpretability” part, the interpretability can be incorporated by simply adopting the decoder in TimeVAE in our framework for continuous only signals. Interpretability of time series data has a long history, and it is deeply rooted in the assumption that the signal is decomposable into three parts: trend, seasonality, and noise. But the main bottleneck is “it is not clear on how to define the decomposability of categorical data”, which requires further research and notions. Given this, this topic is out of the scope of our current work and defers it to future research. We would like the reviewer to view our work as a trial for modeling time series tabular data with heterogeneity and show how the model can be used in time varying metadata conditional generation. (Please see our section 4.3 in the newly revised manuscript.)
>
> Q) Another point is the pure focus on continuous data: While the method demonstrates good performance with continuous data, its methods are primarily tailored for this data type. This focus raises concerns about the model's effectiveness when dealing with heterogeneous datasets that include categorical features. The inability to seamlessly integrate and process diverse data types could restrict the model's usability in real-world applications where such heterogeneity is common.
>
> A) Thanks for the comment, but we want to emphasize our model TimeAutoDiff is primarily designed for taking care of heterogeneous features, which both include continuous and categorical features. The key idea is to combine VAE and diffusion model for dealing with this heterogeneity, and we design our own VAE and diffusion to encode the temporal dependencies of rows in the data by using the inductive bias of RNNs. The dataset used in the experimental parts have heterogeneous features, and by the suggested four metrics in our paper, our model gives the superior performance when compared with other time series data synthesizers, which didn’t consider modeling the heterogeneity of data.
>
> Q) Another point is the dependence on latent space reduction: The paper notes a performance drop when the feature sizes increase, which necessitated a reduction in the latent space dimension. This reliance on dimensionality reduction to maintain performance raises concerns about the model's scalability and robustness. If the model's effectiveness is sensitive to the dimensionality of the latent space, it may struggle to perform well in high-dimensional settings without careful tuning.
>
> A) Latent space dimensionality reduction should not be viewed as a disadvantage of the model; rather, it is a distinct merit that provides opportunities to enhance performance and efficiency in the inference step of diffusion model, and it is an aspect we aim to leverage effectively in our problem setting, just in case we have features in high-dimensional space.
> The primary purpose of latent diffusion model in vision-related tasks is to speed up the sampling time. For instance, see Vadhat’s paper. They project the image data in a high-dimensional original data space to low-dimensional latent space and let the diffusion model learn the projected data in low-dimensional space. Since, diffusion model is trained on low-dimensional space, the sampling time in diffusion becomes much shorter, while maintaining the good quality of generated images. (Here, we assume the decoder in VAE is expressive enough.)

---

> > ### Author Response · Authors · 2024-11-24
> > **Response to the comments from Reviewer Q7zx (2)**
> >
> > (Continued answer for your previous comments.)
> >
> > We would like to note the reviewer that the main reason why we set the dimension of latent space same with that of the original data space in our main experiments (Table 1)  is because most time series data available to the public has small number of features, (See the Table 6 in the Appendix B, the dataset with the largest feature is 28: Energy dataset.). Since the feature dimension in the original space is not that big, we didn’t bother ourselves with the dimension reduction in the feature dimension. But indeed, when feature dimension is in the range of 30~50, we observe a performance drop both in fidelity and sampling time. The main reason for the performance drop in fidelity is because the VAE’s reconstruction error increases significantly, and main reason for having longer sampling time is from diffusion model.
> >
> > $\textbf{Questions:}$
> >
> > 1, How can the model be made more robust to high-dimensional feature spaces without relying heavily on latent space reduction?
> >
> > Answering your first question, a possible way of improving the robustness of the model is to increase the size of VAE, so that it can be expressive enough to accommodate the data in the high-dimensional feature space. But we don’t recommend this way because of the curse of dimensionality, we need to increase the size of the network exponentially in feature dimension. Instead
> >
> > 2, What modifications or extensions to the framework would be necessary to effectively handle heterogeneous datasets that include both continuous and categorical features?
> >
> > A)	Please see our answer to the second reviewer’s second comments (soundness) for the idea on how to combine VAE and Diffusion model
> > B)	Please see the caption of Fig 1. in our manuscript.
> > C)	Please see the answer to the second comment of yours in Weakness section.

---

### Official Review · Reviewer_AwgG · 2024-11-04

**Soundness:** 3
**Presentation:** 3
**Contribution:** 2
**Rating:** 3
**Confidence:** 2

**Summary:**

The proposed work presents a framework of latent diffusion models for heterogeneous time series data. The framework utilizes an encoder within a variational autoencoder (VAE) to project both discrete and continuous features of time series data into a continuous latent space. Subsequently, a diffusion model is employed to model the distribution of the latent codes of the time series data. The encoder employed in the VAE is an RNN, while the diffusion model is DDPM[1]. Notably, the denoising network within DDPM employs cyclic encoding to incorporate time-stamp information into the time series and adopts a bi-directional RNN architecture. The proposed latent diffusion model can also be conditioned on meta-data for incorporating conditional information. Evaluation results on unconditional generation demonstrate that the proposed approach generates high-quality synthetic data and outperforms existing approaches.

References

[1] Ho, Jonathan, Ajay Jain, and Pieter Abbeel. “Denoising diffusion probabilistic models.” Advances in neural information processing systems 33 (2020): 6840-6851.

**Strengths:**

The work has the following notable strengths:
1. The works did comprehensive evaluation on unconditional and conditional generation tasks, taking both lower-order and higher-order statistics into account. The proposed approach shows superior performance over existing approaches on both tasks across the datasets considered in the work. In addition the work also did comprehensive over different hyper-parameters variations and ablations to reveal the impact of different components to the model.
2. The presentation quality of the work is good and the proposed method of the work is easy to follow.

**Weaknesses:**

The work exhibits several significant weaknesses:
1. The work lacks innovative methodology. Similar ideas for tabular data, which arguably encompass time-series data, have been explored in existing works [1]. Despite some crucial technical differences between the two works, including the design of the denoising network $\epsilon_\theta$ and the approach to the latent space, the high-level framework of both works is both a latent diffusion model that projects structured data to a latent space and uses a diffusion model to model the distribution of latent codes.
2. The majority of the experiments are limited to unconditional generation of time-series data, and it is unclear how the proposed model can be readily adapted to tasks with more practical applications, such as forecasting and imputation.

References

[1] Zhang, Hengrui, et al. “Mixed-type tabular data synthesis with score-based diffusion in latent space.” arXiv preprint arXiv:2310.09656 (2023).

**Questions:**

1. I would like to inquire about the rationale behind employing RNN as the encoder for time-series data and capturing temporal dependencies between features across various timestamps. The temporal dependencies between features can also be captured by the denoising network of the diffusion model. Therefore, is it necessary to utilize an RNN to capture temporal dependencies when encoding the data?

2. I would encourage the author to provide a more comprehensive discussion and emphasize the values or practical significance of the tasks the model is evaluated on, specifically unconditional generation and time-variant meta-data conditioned generation.

---

> ### Author Response · Authors · 2024-11-24
> **Response to the comments from Reviewer AwgG (1)**
>
> 1.	The work lacks innovative methodology. Similar ideas for tabular data, which arguably encompass time-series data, have been explored in existing works [1]. Despite some crucial technical differences between the two works, including the design of the denoising network ϵθ and the approach to the latent space, the high-level framework of both works is both a latent diffusion model that projects structured data to a latent space and uses a diffusion model to model the distribution of latent codes.
>
> A)	Thanks for your comment. This is somehow duplicate answer to the second reviewer’s first comment. But we don’t think adopting a Latent diffusion model framework is a significant drawback in terms of novelty of our work. LDM (VAE + Diffusion) is already a widely used framework in various domains other than tabular domain: DALLE-2 (text-to-image model from OpenAI), GeoLDM (3D-Moleculer generation), VideoLDM (video generation). And we borrowed this idea in the tabular domain (which is rarely studied) to tackle the heterogeneity issue and showed it can indeed solve this issue in time series modeling. (Please see the “Motivation of TimeAutoDiff” section in Subsection 1.1. in our newly revised manuscript.) We believe this is a useful / helpful idea to the community and believe this can be further expanded to solve more practically involved real-world problems as we mentioned in the conclusion part.
>
> B)	As mentioned in our paper, TimeAutoDiff is inspired by TabSyn [1] and AutoDiff [2], employing LDM framework. But we argue that TabSyn [1] cannot encompass the time-series data. The time series data has 2-dimensional form (i.e., 2D data): one dimension for time-axis and another dimension for feature axis, whereas the data format TabSyn can afford is in 1-dimensional space (only feature dimension). In their modeling scheme, they take each row of the table as one data point, and think the rows are i.i.d., and this means they didn’t model the temporal dependencies of rows in the table, which is an obvious difference between our work and theirs.
>
> 2.	Most of the experiments are limited to unconditional generation of time-series data, and it is unclear how the proposed model can be readily adapted to tasks with more practical applications, such as forecasting and imputation.
>
> A)	Thanks for the comment. We take your comments seriously, and felt indeed the original version lacks the part how our model can be used in practice. Both imputation and forecasting tasks can be framed as conditional generation, and they can be tackled through masking techniques. But we are planning these tasks as separate projects to deal with these topics in a more in-depth manner. (We also give the detailed discussion section for this topic in Appendix J.)
>
> B)	Instead, we work on meta-data conditioned generation to show how our model can be used in practical applications. Taking your comments seriously, we moved the scalability and ablation tests parts to the Appendix and move the time varying metadata part that was originally in the Appendix M to the main body of the revised manuscript. In this section, under synthetic and real-data setting, how our model can be used for modeling P(Continuous | Discrete) and P(Discrete | Continuous) distribution. We believe this functionality is useful for counterfactual scenario explorations in many fields, given the prevalence of heterogeneous time series data in the real-world.
> References
>
> [1] Zhang, Hengrui, et al. “Mixed-type tabular data synthesis with score-based diffusion in latent space.” arXiv preprint arXiv:2310.09656 (2023). \
> [2] Namjoon, Suh et al. "AutoDiff: combining Auto-encoder and Diffusion model for tabular data synthesizing." NeuRIPS 2023, SyntheticData4ML workshop.

---

> ### Author Response · Authors · 2024-11-24
> **Response to the comments from Reviewer AwgG (2)**
>
> $\textbf{Questions:}$
>
> 1.	I would like to inquire about the rationale behind employing RNN as the encoder for time-series data and capturing temporal dependencies between features across various timestamps. The temporal dependencies between features can also be captured by the denoising network of the diffusion model. Therefore, is it necessary to utilize an RNN to capture temporal dependencies when encoding the data?
>
> A)	Nice catch. Followings are answers to your questions: \
> \
> $\textbf{Hierarchical Temporal Dependency Modeling:}$ The VAE encoder is designed to capture a compact representation of the input data, including temporal dependencies, which serves as the latent variable for the subsequent diffusion process. Encoding temporal dependencies explicitly in the latent space helps the diffusion model operate on a representation that already integrates key temporal patterns. This hierarchical approach allows the diffusion model to focus on modeling finer-grained dependencies and generating realistic outputs, rather than redundantly encoding high-level temporal structures.
>
> $\textbf{Noise-Tolerant Latent Representation:}$ By capturing temporal dependencies early in the VAE encoder, the latent variable $\mathbf{z}$ becomes a noise-tolerant representation of the time-series data. This benefits the diffusion process, as the added noise during diffusion is less likely to obscure important temporal structures, improving the overall quality and fidelity of generated data.
>
> Inspired by your comments, we performed additional ablation tests. We replaced the RNN architecture in our encoder with fully connected layers and observed the performance of the ablated model at the sub-par level when compared with the original configuration. Please check the table 7 in the Appendix L.
>
> 2.	I would encourage the author to provide a more comprehensive discussion and emphasize the values or practical significance of the tasks the model is evaluated on, specifically unconditional generation and time-variant meta-data conditioned generation.
>
> A)	Please see our answer to your second comments on weakness.

---

> > ### Comment · Reviewer_AwgG · 2024-11-25
> > **Response to Rebuttal**
> >
> > Thank you for the detailed response and additional experiment results. I appreciate the author's the clarification on the critical difference between tabular data and heterogeneous time series data. However, I'm worried about the contributions of the work when considering the author's overall response to both of my concerns. Adapting existing LDM framework to time-series data and showing results mostly on unconditional generation of little practical values is not a significant enough contribution for publication on ICLR. My suggestion to the author is to include experiment results on some basic tasks in time series like forecasting and imputation and I don't think it should be a separate work as many existing works [1, 2, 3] applying diffusion models to time series already studied these tasks. The author also mentioned that meta-data conditioned generation could be useful for counterfactual scenario which is related to OOD or anomaly detection. This could be another potentially significant contribution of the work but it is not well supported by experiment results in the current version.
> >
> > References
> >
> > [1] Biloš, Marin, et al. "Modeling temporal data as continuous functions with process diffusion." (2022).
> >
> > [2] Tashiro, Yusuke, et al. "Csdi: Conditional score-based diffusion models for probabilistic time series imputation." (2021).
> >
> > [3] Shen, Lifeng, Weiyu Chen, and James Kwok. "Multi-Resolution Diffusion Models for Time Series Forecasting." (2024).

---

> > > ### Author Response · Authors · 2024-11-25
> > > **Thanks for your comments**
> > >
> > > We would like to emphasize that the three papers you referenced focus on continuous time series modeling. In contrast, our work addresses an additional layer of complexity: heterogeneous features. This heterogeneity, combined with the temporal dependencies in tabular time series data, introduces unique challenges that are the central focus of our study.
> > >
> > > We acknowledge that unconditional generation may not have as many immediate practical applications compared to forecasting or imputation tasks. However, we believe that modeling the full data distribution is a crucial foundational step. A robust understanding of the data’s underlying distribution provides a solid basis for extending to tasks like forecasting and imputation, which can naturally be framed as conditional generation with appropriately designed masks. Addressing these tasks in the context of heterogeneous features is non-trivial and requires deeper exploration beyond straightforward adaptations of existing methods.
> > >
> > > Regarding your suggestion to explore meta-data conditioned generation for counterfactual scenarios, we greatly value this feedback and see its potential for enhancing our work. In Section 4.3, we have taken preliminary steps in this direction, but we recognize the need for further refinement. If you have specific suggestions or insights on how we could improve these experimental results, we would greatly appreciate your input. It would be good to make our work published, but we think making the better paper is more important. It would be immensely helpful to hear your thoughts on how to make this aspect of the work more compelling.
> > >
> > > Thank you again for your constructive feedback and for highlighting opportunities to improve our work. We remain open to further dialogue and are committed to addressing these points in a meaningful way.

---

### Official Review · Reviewer_KEcj · 2024-11-04

**Soundness:** 1
**Presentation:** 1
**Contribution:** 1
**Rating:** 1
**Confidence:** 1

**Summary:**

This paper proposes to use latent diffusion models to generate synthetic time series tabular data.
They propose to combine VAE with DDPM and term the proposed method TimeAutoDiff.
They claim many advantages of this proposal and provide some experiments.

**Strengths:**

NA

**Weaknesses:**

**Originality.** DDPM or VAE for time series generation is not new. Most techniques used are also from prior works.
**Soundness.** It’s well-known that VAE is considered 1-step DDPM. I find the motivation and reasoning in `line 144-154` very unconvincing. It’s still unclear why DDPM is good at handling time series data, why VAE, as a special case is DDPM, can bring more to the table. Also, I don’t think citing unpublished and prior venue’s rejections as SOTA method is convincing, e.g., TSGM in `line 156`.
**Clarity.** The captions of Fig 2,3 provide zero information. Many notations are introduced without definition, e.g., x_cont in `line 192`. Overall clarity can be further improved.
**Significance.**  The significance is hard to parse due to limited clarity. At best, I find it lacking due to its assembly nature. Additionally, I question the correctness of the experimental evaluation.

**Questions:**

Why are most experiments on regression datasets? How can these validate heterogeneous properties of “tabular” time series data?

The experiments lack convincing evidence. Please include additional summary statistics of the generated data and compare with baselines. For instance, stock data (nasdaq100) are known for key summary statistics like volatility and moving averages. Demonstrating that the proposed method closely matches these statistics would provide stronger support for its efficacy. Can you provide a generated example of your trained nasdaq100 model?

Why is Section 4.2 titled “Utility Guarantees” when there is no theoretical analysis provided?

These weaknesses are not meant to be exhaustive. I believe they are sufficient to show that this paper is not ready for publication.

---

> ### Author Response · Authors · 2024-11-24
> **Response to the comments from Reviewer KEcj (1)**
>
> Thanks for your comments on our work! We made significant revisions inspired by your feedback. Please take a look at our revised manuscript and followings are the answers to your questions and comments on weakness and questions:
>
> Originality. DDPM or VAE for time series generation is not new. Most techniques used are also from prior works.
>
> A) Both models are widely used in generative modeling, and we believe it is effective to leverage the strengths of established methods to address certain problems, rather than introducing complex new approaches. We hope the reviewer recognizes our work as an effort to tackle tabular time series modeling with heterogeneous features by effectively applying two powerful existing models. Furthermore, latent diffusion model (VAE + Diffusion) is already a widely used framework in various domains other than tabular domain: Dalle-2 (text-to-image model from OpenAI), GeoLDM (3D-Moleculer generation), VideoLDM (video generation). And we borrowed this idea in the tabular domain (which is rarely studied) to tackle the heterogeneity issue and showed it can indeed solve this issue in time series modeling. (Please see the “Motivation of TimeAutoDiff” section in Subsection 1.1. in our newly revised manuscript.) We believe this is a useful / helpful idea to the community and believe this can be further expanded to solve more practically involved real-world problems as we mentioned in the conclusion part.
>
> Soundness. It’s well-known that VAE is considered 1-step DDPM. I find the motivation and reasoning in line 144-154 very unconvincing. It’s still unclear why DDPM is good at handling time series data, why VAE, as a special case is DDPM, can bring more to the table. Also, I don’t think citing unpublished and prior venue’s rejections as SOTA method is convincing, e.g., TSGM in line 156.
>
> A) VAE is a useful tool for handling heterogeneity of tabular data. In regular tabular data setting (i.i.d row setting), (for instance, TabSyn, AutoDiff) VAE or autoencoder framework have already been widely used. We employ this idea into time series tabular setting with our own encoder & decoder in VAE. The handling of heterogenous feature is easy through VAE because VAE’s objective function is involved with the reconstruction error, whereas diffusion model is primarily designed to minimize the negative log-likelihood function of continuous random variables. There’s no way we can deal with heterogeneity of data so far with only one diffusion model, other than combining two different types of diffusion models for continuous and discrete variables (for instance, in tabular modeling literature: TabDDPM, TimeDiff). But this method is known for not being able to capture the correlations among the continuous and discrete random variables.
>
> Regarding your comments on diffusion model, you are right. Diffusion model can be thought of as VAE with T (diffusion steps) hidden layers in both encoder and decoder. (Using score-based framework, it becomes infinite layers.) These multiple diffusion steps (hidden layers) enable iterative refinements of distributional modelling. In the forward process, the prior distribution gets close to standard Gaussian, nonetheless, VAE (arguably with one diffusion step diffusion model) lacks this expressive power. You can recall the tradeoff between reconstruction error and KL-divergence in VAE, and if we harshly penalize the divergence, restricting the shape of latent space to follow Gaussian, the reconstruction will be very high, and vice versa. But diffusion model can easily avoid this problem, enabling the distributional modeling for high-dimensional problems. (In our case, the data dimension is $T\times F$).
>
> Why diffusion model is good for time series data modeling? Diffusion models are particularly well-suited for time series modeling due to their ability to handle the complexities of sequential data while capturing the inherent dependencies and uncertainties. The iterative denoising process of diffusion models allows them to effectively learn temporal dependencies across time points, making them ideal for modeling the sequential dynamics in time series. Their flexibility enables the integration of specialized architectures, such as RNNs or Transformers, within the denoising network to better capture temporal relationships. Furthermore, diffusion models excel at handling heterogeneous data, which is common in time series, by leveraging tailored output heads and loss functions (e.g., MSE, BCE, CE) for mixed feature spaces. The noise injection and removal mechanism of diffusion models enhances robustness, allowing them to manage stochastic elements, outliers, and irregularities in the data. Additionally, diffusion models are designed to scale well with high-dimensional inputs, making them suitable for complex multivariate time series data with long horizons.

---

> > ### Author Response · Authors · 2024-11-24
> > **Response to the comments from Reviewer KEcj (2)**
> >
> > (Continued answer for your previous comments.)
> >
> > Another significant advantage of diffusion models is their ability to support conditional generation. They can be conditioned on metadata such as timestamps or external variables, enabling tasks like forecasting, counterfactual analysis, and imputation. As generative models, they are also effective at synthesizing realistic time series, which is useful for data augmentation or simulation. The probabilistic framework of diffusion models provides a natural way to represent uncertainty and variability in time series, while their iterative denoising process offers interpretability, allowing insights into how temporal structures are reconstructed from noise. Moreover, their versatility makes them applicable to a wide range of time series tasks, including forecasting, anomaly detection, and missing data imputation. Overall, diffusion models combine robustness, flexibility, and generative power, making them a compelling choice for addressing the challenges of time series modeling.
> >
> > Regarding your comment on the comparison of our model with TSGM: currently, we are not aware of many timeseries synthesizers (specifically, diffusion-based synthesizers) whose codes are publicly available, and fair enough to say their problem settings are same with ours. Furthermore, some models whose codes are released to the public; their codes didn’t work properly. TSGM was one who satisfied the above criteria.
> >
> > GeoLDM: Geometric latent diffusion models for 3d molecule generation, M. Xu et al., 2023, ICML \
> > VideoLDM: Align your latents: High-resolution video synthesis with latent diffusion models, Blattmann et al., 2023, CVPR
> >
> > $\textbf{Clarity}$. The captions of Fig 2,3 provide zero information. Many notations are introduced without definition, e.g., x_cont in line 192. Overall clarity can be further improved.
> >
> > A) Thanks for the comments. We try to make detailed descriptions on Fig 2 and 3. And try to give explanations / descriptions on every notation used in the paper. Specifically, we revised significantly the VAE part in our manuscript.
> >
> > $\textbf{Significance.}$ The significance is hard to parse due to limited clarity. At best, I find it lacking due to its assembly nature. Additionally, I question the correctness of the experimental evaluation.
> >
> > A) We are proud of our work, and we think our work will provide valuable contributions to the synthetic time series generation community. Please take one more look at our revised work and hope the revisions we made make sense to you. Appreciate for your valuable comments.
> >
> > $\textbf{Questions:}$
> >
> > Q) Why are most experiments on regression datasets? How can these validate heterogeneous properties of “tabular” time series data?
> >
> > A) Admittedly, we did not include classification datasets in our experiments, as our focus was on measuring predictive scores—a regression task—to evaluate the performance of the generated data in downstream tasks. The predictive score serves a critical purpose: to determine whether a model trained on synthetic data can generalize effectively to real data. Importantly, this task inherently validates the utility of the heterogeneous features in the synthetic data.
> > In our experiments, the regression setup ensures that the model predicts $y_{t}$ (a chosen target variable, such as the last variable in the dataset) using $X_{t−1}$ , which includes all remaining heterogeneous features at the $(t−1)$-th timestamp. This formulation ensures that the synthetic data preserves the complex interdependencies among heterogeneous features, as the model relies on these features to make accurate predictions. If the synthetic data fails to capture the statistical and temporal relationships between the features, the predictive performance on real data would degrade significantly. Thus, the regression task effectively serves as a validation of the proper generation of heterogeneous features in the synthetic data.
> > Moreover, the regression approach evaluates both the temporal coherence and the feature relationships, making it a robust indicator of the synthetic data's ability to replicate the characteristics of tabular time series data. While classification tasks could also serve as a validation method, regression tasks provide a comprehensive framework for assessing the fidelity of heterogeneous features and their interdependencies in synthetic data.
> > Other than predictive score, we additionally give the validity of our model for generating heterogeneous data, we consider: (1) the discriminative score and temporal discriminative score, which assesses the fidelity of the generated data by comparing it to the real data; (2) feature correlation, which measures whether the generated data preserves the relationships among features, particularly important for heterogeneous features. Additionally, in the Appendix O, Maximum Mean Discrepancy (MMD) and Entropy to measure the similarities and diversity of the generated heterogeneous data.

---

> > > ### Author Response · Authors · 2024-11-24
> > > **Response to the comments from Reviewer KEcj (3)**
> > >
> > > Q. The experiments lack convincing evidence. Please include additional summary statistics of the generated data and compare with baselines. For instance, stock data (nasdaq100) are known for key summary statistics like volatility and moving averages. Demonstrating that the proposed method closely matches these statistics would provide stronger support for its efficacy. Can you provide a generated example of your trained nasdaq100 model?
> > >
> > > A. Thank you for your suggestion. In the financial domain, volatility and moving average is a critical metric that synthetic data must accurately capture.
> > > In the revised paper, the Figure 10 provide a clear side-by-side comparison between the synthetic and real data, with the left column displaying the synthetic data and the right column showcasing the corresponding real data for two selected features (Open \& Close prices) over 200 timestamps (i.e.,T=200). Each row focuses on one feature, allowing for a detailed examination of the behavior across key metrics: Simple Moving Average (SMA), Exponential Moving Average (EMA), and Volatility. The SMA and EMA curves, plotted alongside the raw time series data, highlight the ability of the synthetic data to replicate the long-term trends (SMA) and short-term responsiveness (EMA) observed in the real data. Volatility, overlaid as a secondary y-axis in each plot, demonstrates the synthetic data's capacity to reproduce the temporal variability, including periods of high and low uncertainty, as reflected in the real data. The remarkable alignment across all metrics suggests that the synthetic data closely mirrors the real data’s dynamics, effectively capturing both the overall patterns and nuanced fluctuations. This visual comparison underscores the robustness and reliability of the synthetic data generation process. We provide the code for reproducing this result in the new supplementary link (File name: Unconditional Generation). Feel free to run the code. And we also provide our generated data. We didn't include the results from the baseline methods because we think in order to measure the volatility reasonably well, we need long enough sequence length, but the baseline method cannot afford that.
> > >
> > > Q. Why is Section 4.2 titled “Utility Guarantees” when there is no theoretical analysis provided?
> > >
> > > A. What we mean originally by the Guarantees is “Empirical Guarantees”. We remove the word “Guarantees” and change the title to “Fidelity and Utility of Synthetic Data.” to avoid the confusion.

---

> > > > ### Author Response · Authors · 2024-11-24
> > > > **Response to the comments from Reviewer KEcj (4)**
> > > >
> > > > We just uploaded a new zipped file for our code.
> > > > Please run jupyter notebook titled: Unconditional Gen (Submission) for checking the volatility and moving average of our synthetic data. We also upload the synthesized data: titled: stock_data_for_volatility_synthetic.pt
> > > >
> > > > Thanks for your comments!

---

### Official Review · Reviewer_98zq · 2024-11-04

**Soundness:** 3
**Presentation:** 3
**Contribution:** 3
**Rating:** 5
**Confidence:** 3

**Summary:**

The paper proposes a novel method, TimeAutoDiff, to generate time series. It combines a variation autoencoder and DDPM. The method empirically outperforms prior methods on several datasets and allows for conditional time series generation.

**Strengths:**

- **Writing** The paper is well-written and well-structured.
- **Methodological contribution** The paper presents a clear method similar to diffusion methods in other domains. The technical details are discussed in detail. It seems straightforward to reproduce the results.
- **Research problem** The presented problem is essential and timely. In particular, synthetic time series generation is critical for foundation model development, as the authors mention in the discussion.

**Weaknesses:**

See Questions for detailed comments and suggestions.

- **Comparison to prior methods** It is worth discussing more the key differences to other diffusion-based models such as Diffusion-ts (https://github.com/Y-debug-sys/Diffusion-TS), and TimeDiff (https://openreview.net/pdf?id=ESSqkWnApz). Additionally, it is worth expanding metrics, such as consistency and diversity, as discussed in the TSGM framework (https://github.com/AlexanderVNikitin/tsgm).

- **Limitations and fairness**. Limitations and fairness are not discussed enough. In particular, can generated data be biased with respect to conditional metadata?

- **Design choices**. It would be great to have more details on the design choices of the decoder and $\epsilon_\theta$. Do other architectures significantly affect the performance of the method?

**Questions:**

>The sampling time column ranks models by their speed, with lower numbers indicating
faster sampling

Please add details.


> We set the dimensions of output features in this way as we used
the mean-squared (MSE), binary cross entropy (BCE), and cross-entropy (CE) in the Pytorch package.

It looks disconnected from the main text. Could you clarify?

>L: 249 More details are deferred in the Appendix J.

Appendix J does not provide additional details on the selection of a decoder architecture.

> LL: 171-173

Does the current model generate data for textual metadata? If so, it would be great to provide more experiments/demonstrations. If not – it may be worth removing this example.

---

> ### Author Response · Authors · 2024-11-24
> **Response to the comments from Reviewer 98zq**
>
> Thanks for your comments on our work! We made significant revisions inspired by your feedback. Please take a look at our revised manuscript and followings are the answers to your questions and comments on weakness and questions:
>
> •	Comparison to prior methods : It is worth discussing more the key differences to other diffusion-based models such as Diffusion-TS (https://github.com/Y-debug-sys/Diffusion-TS), and TimeDiff (https://openreview.net/pdf?id=ESSqkWnApz). Additionally, it is worth expanding metrics, such as consistency and diversity, as discussed in the TSGM framework (https://github.com/AlexanderVNikitin/tsgm).
>
> A.)	Diffusion-TS’s main purpose is to generate time series data with interpretability. They employ the Autoencoder + DDPM framework, employing transformers as encoder and decoder for obtaining the disentangled representations of time series.
>
> The main difference between Diffusion-TS and ours is on the problem setting that their assumption on the signal is only restricted to continuous time series, whereas ours is focused on the heterogeneous features. Diffusion-TS lies on the assumption that the signal is decomposable into three main parts: trend, seasonality, and noise. However, the decomposition of heterogeneous features, specifically discrete variables is not well defined in the literature, it is beyond the scope of our work, requiring further research.
>
> TimeDiff integrates two types of diffusion models to handle heterogeneous features in EHR datasets, employing DDPM for continuous variables and multinomial diffusion (Hoogeboom et al., 2021) for discrete variables. In contrast, our approach leverages a VAE to project time series data into a latent space and utilizes DDPM exclusively for modeling the time series within this latent representation, which is continuous. We include this comparison in the Appendix E.
>
> B.)	Appreciate for the comments. After carefully reading the TSGM paper, we realized the consistency metric they proposed is for measuring the downstream performance of classification tasks. Nonetheless, to be computed, this metric is restrictive requiring several conditions that is not compatible with our original experimental setting. First, the data should be the classification time series dataset; that is, for each multivariate time series data, they should have the label. But the dataset we consider in this paper is not designed for classification problems. Second, the metric requires the data to be generated from conditional model. But for comparison purposes, the baseline models that we employed cannot afford the conditional generation, except for our model. Instead, we measure the predictive score which is the most widely used metric for measuring the performance of synthetic data in downstream tasks in the literature (See Table 1 in our paper.).
>
> Nonetheless, we used two metrics suggested by TSGM: MMD and Entropy. Maximum Mean Discrepancy (MMD) is for measuring the similarities between synthetic and real time series data, and Entropy measures the diversity of the synthetic data. The results are summarized in the Table 11 in the newly revised manuscript. (Appendix O)
>
> We used two metrics proposed by TSGM [nikitin2023 et al]: Maximum Mean Discrepancy (MMD) and Entropy. MMD measures the similarity (or fidelity) between synthetic and real time series data, while Entropy assesses the diversity of the synthetic data. The results are summarized in Table 11 and are consistent with those in Table 1.
>
> TimeAutoDiff achieves the lowest MMD scores across all four datasets, aligning with the discriminative scores reported in Table1. This indicates that TimeAutoDiff effectively generates synthetic data that closely resembles real data.
> For diversity, higher Entropy values indicate a dataset with more diverse samples. However, as noted in TSGM, Entropy should be considered alongside other metrics, as random noise can also result in high Entropy values. TimeAutoDiff produces synthetic data with higher Entropy values than the real data, though not as excessively as other baseline models. This suggests that our model generates synthetic data that preserves the statistical properties of the original data, maintaining diversity without introducing excessive deviation.

---

> ### Author Response · Authors · 2024-11-24
> **Response to the comments from Reviewer 98zq (2)**
>
> •	Limitations and fairness: Limitations and fairness are not discussed enough. In particular can generated data be biased with respect to conditional metadata?
>
> Generated data can indeed be biased with respect to conditional metadata, arising from various factors. Bias in the training data, such as inherent associations between metadata and outputs, may lead the model to replicate these biases, for instance, generating disproportionately high traffic volumes for "Clear" weather even when the true relationship is less deterministic. Imbalanced metadata distributions further exacerbate this issue, as underrepresented conditions in the training set often result in less reliable outputs for those conditions, such as biased outcomes for minority demographic groups in healthcare datasets. Simplified assumptions in the model, such as assuming linear relationships between metadata and outputs, can overlook complex dependencies, producing data that fails to reflect the true conditional distribution. Noise injection, a feature of models like diffusion models and VAEs, can introduce additional bias if the noise interacts with metadata in unexpected ways, particularly for rare metadata values. Furthermore, limitations in conditional architectures, such as inadequate metadata encoding, can prevent the model from capturing nuanced dependencies, leading to misaligned outputs. To mitigate such biases, ensuring balanced training data, employing robust metadata encoding techniques, applying regularization or fairness constraints, performing post-generation bias audits, and designing disentangled latent spaces are crucial steps. While conditional generative models aim to align generated data with metadata, addressing these biases is essential to ensure fairness and reliability. We include this paragraph in the Appendix A of our revised manuscript.
>
> •	Design choices: It would be great to have more details on the design choices of the decoder and ϵθ. Do other architectures significantly affect the performance of the method?
>
> (A). We add detailed descriptions on our decoder in Appendix E.
> (B). We add several ablation tests inspired by your comments. We replaced MLP with the RNN structure in the decoder.
> (C). Inspired by [Bilos et al., 2023], we explore injecting continuous noise from a stochastic process (Gaussian process) into the DDPM.
> The experimental results show that none of the ablated models outperformed the original configuration significantly. Specifically, the second configuration demonstrates the benefits of modeling temporal relations twice in the VAE and DDPM due to the following reasons:
>
> $\textbf{Hierarchical Temporal Dependency Modeling:}$ The VAE encoder captures compact latent representations with temporal dependencies, providing a structured input for the diffusion model. This allows the diffusion process to refine finer-grained patterns without redundantly encoding high-level temporal structures, resulting in more realistic outputs.
>
> $\textbf{Noise-Tolerant Latent Representation:}$ Encoding temporal dependencies early in the VAE encoder ensures that the latent variable $\mathbf{z}$ is robust to noise. This noise resilience helps maintain critical temporal structures during the diffusion process, enhancing the fidelity of the generated data.
>
> Bilos et al., 2023 ICML: Modeling temporal data as continuous functions with stochastic process diffusion
>
> $\textit{Questions:}$
>
> Q. The sampling time column ranks models by their speed, with lower numbers indicating faster sampling. Please add details.
>
> A. The detailed sampling time (in second) is in the last row of Table 1. Additionally, we move the table to Appendix E.
>
> Q. We set the dimensions of output features in this way as we used the mean-squared (MSE), binary cross entropy (BCE), and cross-entropy (CE) in the Pytorch package. It looks disconnected from the main text. Could you clarify?
>
> A. We mentioned this because the BCE and CE functions in the PyTorch package take specific arguments for input and target dimensions. Specifically:
>
> •	BCE takes one argument: the logit (unnormalized score) for each class when the label is equal to 1. This requires the input to have the same dimensions as the target (e.g., [batch_size, num_features]), where values represent logits for each feature.
>
> •	CE expects the input to be the logits (unnormalized scores) with shape [batch_size, num_classes] and the target to be class indices with shape [batch_size] (not one-hot encoded).
>
> •	MSE requires both the input and target tensors to have the same shape (e.g., [batch_size, num_features]), where values represent continuous outputs.
>
> To ensure compatibility with these loss functions, we carefully set the dimensions of the output features to align with the expected input shapes for MSE, BCE, and CE during training. We revised the paragraph in our newly revised paragraph. [Line 240~247.]. We provide the figure of our decoder in the Appendix L, Figure 12. in our revised manuscript.

---

> > ### Author Response · Authors · 2024-11-24
> > **Response to the comments from Reviewer 98zq (3)**
> >
> > Q. L: 249 More details are deferred in Appendix J.
> > Appendix J does not provide additional details on the selection of a decoder architecture.
> >
> > A. Thanks for the comments! We provide the detailed pictorial description on the decoder architecture in Appendix L. And we also revised the paragraph on decoder part in our main manuscript as well.
> >
> > Q. LL: 171-173. Does the current model generate data for textual metadata? If so, it would be great to provide more experiments/demonstrations. If not – it may be worth removing this example.
> >
> > A.The model does not generate the text data, and we admit this can cause confusion to the reader, so we removed the statement, and we significantly revised the entire paragraph on introducing the metadata conditional generation. Please take a look at Subsection 4.3 in our newly revised manuscript.

---

### Author Response · Authors · 2024-11-24
**Summary of Revision**

Dear Reviewers and AC

First of all, we greatly appreciate for the insightful comments from reviewers on our work. We re-upload the revised version of our manuscript, and all the revised parts are colored in red. Followings are the main summary of the revised manuscript.

1. From reviewer 2's comment, we revise the motivation of VAE + DDPM in section 1.1. We think this is the most important part of our entire section. Hopefully, the revised version makes sense.
2. From multiple reviewers' comments, we feel the real applications of our work are not strong enough. So, we move the demonstration of conditional generation part, which was originally in the Appendix to the main manuscript.
3. We add additional ablation tests on the VAE and DDPM, inspired from the first and third reviewers. The results are added in the Appendix L.
4. We provide additional experiments on volatility, moving average (on stock data) (Appendix J) and MMD, diversity (Appendix O) to further corroborate the good fidelity of synthetic data from our model.
5. Inspired from first reviewer's comment, we add the comments for the bias from conditional metatdata in the Appendix A.
6. We provide the illustrations of decoder in the Appendix L inspired from Reviewer 1's comments.

Thank you all for the valuable suggestions, again! Please let us know if you have additional questions. Thank you,

Authors

---

### Note · Authors · 2024-12-22

I have read and agree with the venue's withdrawal policy on behalf of myself and my co-authors.